# Causal Representation Learning and Inference for Generalizable Cross-Domain Predictions

## Abstract

Learning generalizable representations for machine learning and computer vision tasks is an active area of research. Typically, methods utilize data from multiple domains and seek to transfer the invariant representations to new and unseen domains. This paper proposes to perform causal inference on transportable, invariant interventional distribution to improve the prediction performance under distribution shifts. Specifically, we first introduce a structural causal model (SCM) with latent representations to capture the underlying causal mechanism that underpins the data generation process. Subject to the proposed SCM model, we can perform the intervention on the spurious representations that are affected by domain-specific factors and the latent confounders to eliminate the spurious correlations. Guided by the proposed SCM and the invariant interventional distribution, we propose a causal representation learning framework. Compared to state-of-the-art domain generalization approaches, our method is robust and generalizable under distribution shifts. Furthermore, the empirical study shows that the proposed causal representation scheme outperforms existing causal learning baselines.

## 1 Introduction

While deep learning models have made significant strides in enhancing performance across various applications, from image recognition to natural language processing, they are also known to exhibit shortcomings such as poor generalization in out-of-distribution (OOD) scenarios, a lack of interpretability, and issues related to fairness. These limitations can be attributed to spurious correlations arising from selection biases within the data (Nagarajan et al., 2020), potentially leading to substantial predictive errors. Various efforts have been dedicated to addressing these challenges through domain generalization, transfer learning, explainable AI, and fair AI (Vu & Thai, 2020; Kusner et al., 2017; Blanchard et al., 2021; Arjovsky et al., 2019; Li et al., 2018). However, they have had varied success, often relying on additional assumptions and primarily exploiting correlations that can prove to be spurious. Furthermore, many existing methods aimed at enhancing generalization require data from multiple domains, which can be impractical in real-world applications.

In this study, we propose a novel framework grounded in causal learning. Causal learning employs a structural causal model (SCM) to capture the underlying causal mechanisms that encode intrinsic, stable, and interpretable relationships within the data. Consequently, causal learning exhibits robustness to changes in exogenous factors, invariance under distribution shifts, and strong performance in OOD settings. Current endeavors in causal representation learning for improving OOD prediction can be categorized into two main approaches: 1) learning without intervention and 2) learning with intervention. Methods in the former category include stable representation learning approaches (Cui et al., 2020; Cui & Athey, 2022; Kuang et al., 2018; 2020), as well as invariant representation learning methods across multiple domains (Arjovsky et al., 2019; Koyama & Yamaguchi, 2020; Chalupka et al., 2014; Gao et al., 2021). Stable representation learning typically focuses on learning causal or anti-causal representations of the target variable. Invariant representation learning, on the other hand, seeks to learn a set of invariant features based on specific invariance criteria from data collected across multiple environments. However, these approaches often rely on domain-specific knowledge and work effectively only when a sufficient number of diverse domains is available during training. Additionally, they often assume causal sufficiency, neglecting the pres-

ence of latent confounders, which can significantly limit their accuracy and applicability. Methods in the latter category aim to perform either active and passive intervention in accordance with a predefined or learned SCM (Mao et al., 2021; Liu et al., 2022; Wang et al., 2020; Mao et al., 2022; 2021). Mao et al. (2021) approximate data from different domains by performing interventions based on a predefined SCM, which allows them to generate synthetic interventional data. Subsequently, they carry out invariant representation learning using both observed and interventional data. Liu et al. (2022); Wang et al. (2020); Mao et al. (2022) learn causal representations under interventions and aim to estimate the interventional distribution of causal effects between images and labels through backdoor/frontdoor adjustments. While effective against confounders and domain biases, they often require knowledge of the estimation of confounders and rely on intricate training procedures.

Inspired by these previous works, we introduce a novel framework that learns causal representations and performs causal interventional inference to enhance OOD prediction. Our framework offers three key contributions: 1) We utilize a novel SCM to capture intricate factors and their causal relations that underline the data generation mechanism. 2) Building upon the SCM, We tackle spurious correlations by drawing inferences from interventional distributions, derived specifically from observational distributions. 3) We introduce a training procedure to estimate the essential observational distributions required for computing interventional distributions. To demonstrate the effectiveness of our framework, we conduct experiments on benchmark datasets with distribution shifts. Empirical results showcase the power of our approach, yielding significant improvements over state-of-the-art methods for OOD prediction, while maintaining comparable in-distribution accuracy.

## 2 RELATED WORK

There are two main approaches for learning domain-invariant representations: the causal approach and the non-causal approach.

**Causal approaches:** Causal methods can be categorized by their use of interventions. Methods that do not involve interventions include stable representation learning (Cui et al., 2020; Cui & Athey, 2022; Janzing, 2019; Jiang & Veitch, 2022) and invariant feature learning (Arjovsky et al., 2019; Koyama & Yamaguchi, 2020; Ahuja et al., 2020; 2021b; Rosenfeld et al., 2021; Ahuja et al., 2021a). Stable representation learning methods aim to discover causal or anti-causal features, either through strategies that balance covariates or by incorporating an SCM as a regularization component. Invariant feature learning aims to obtain non-spurious representations by ensuring invariance across environments, emphasizing their robust connections to the target variable. One well-known approach within this realm is invariant risk minimization (IRM). IRM seeks to identify invariant predictors corresponding to the causal parents of a target variable, given multiple environments that correspond to different interventional distributions in a data generation process. Subsequent work has introduced more efficient variants (Ahuja et al., 2020) and conducts further theoretical analyses (Ahuja et al., 2021b). Nevertheless, it has been recently revealed that this principle has limitations in certain scenarios (Rosenfeld et al., 2021; Ahuja et al., 2021a), where it may fail to uncover such predictors. Various other strategies have also been explored, such as risk variance regularization (Krueger et al., 2021), domain gradient alignments (Koyama & Yamaguchi, 2020), smoothing cross-domain interpolation paths (Chuang & Mroueh, 2021), and task-oriented techniques (Zhang et al., 2021). However, these approaches typically either require explicit domain differentiation information or some level of target domain knowledge, making them challenging to apply in real-world scenarios. Causal learning methods involving intervention encompass robust feature learning through data augmentation and transportable interventional inference-guided feature learning techniques. For instance, Mao et al. (2021) performs intervention on input data by identifying a set of transformations that can be applied without compromising invariant features. However, the selection of admissible transformations necessitates domain-specific expertise. Liu et al. (2022), Wang et al. (2020), and Mao et al. (2022) estimate the invariant and transportable interventional distribution between input and target through backdoor/frontdoor adjustments. Nevertheless, these approaches require the identification and estimation of all covariation sources between input and target for backdoor adjustment, limiting their applicability in real-world scenarios. While Mao et al. (2022) avoids this issue by employing front door adjustment, it is computationally expensive to train due to the introduction of integration over the input space.

**Non-causal approaches:** The primary approach is disentangled representation learning. In the early stages, methods focused on enforcing statistical independence among different dimensions of

the learned representation (Achille & Soatto, 2018; Bengio et al., 2013; Burgess et al., 2018; Chen et al., 2018). However, relying solely on statistical independence proved inadequate for achieving effective disentanglement, primarily due to its non-identifiability (Locatello et al., 2019). Recent approaches (Khemakhem et al., 2020; Locatello et al., 2020; 2019; Shen et al., 2022) have sought to enhance disentanglement by incorporating auxiliary information, thereby enabling identifiability and achieving improved results. Other approaches include "mix-up" kind of strategies (Zhang et al., 2017; Yun et al., 2019; Hendrycks et al., 2019), adversarial training strategies (Volpi et al., 2018; Wang et al., 2021) and frequency spectrum strategies (Sun et al., 2021; Zhang et al., 2023). However, these strategies are heuristic and computationally expensive if adversarial training is required.

## 3 CAUSAL INFERENCE FROM INTERVENTIONAL DISTRIBUTION

In this paper, we focus on tasks that perform robust prediction under distribution shifts. We formulate, interpret, and solve such tasks from a causal perspective. We utilize our intuitions regarding the data generation process into an SCM and leverage causal tools to construct invariant, transportable prediction distribution and deduce properties of the ideal representations.

### 3.1 STRUCTURAL CAUSAL MODEL FOR DATA GENERATION PROCESS

Our ultimate objective is to identify the invariant distribution that exhibits generalizability and transportability across various domains. Previous works in causal representation learning (Mao et al., 2021; 2022; Wang et al., 2020; Liu et al., 2022) employ SCMs to describe the data generation process for prediction tasks involving inputs $\boldsymbol{X}$ and targets $Y$. The learning process seeks to capture the underlying causal mechanisms that adhere to a directed acyclic graph (DAG). Building upon and refining the SCMs used in prior methods, we introduce an SCM,

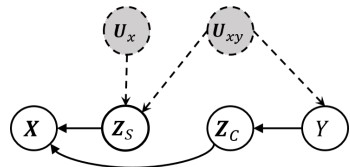

Figure 1: SCM over $(\boldsymbol{X}, \boldsymbol{Z}, Y, \boldsymbol{U})$.

depicted in Figure 1. In this representation, $\boldsymbol{X}$ signifies the high-dimensional input data, such as images, videos, or texts; $Y$ represents the target variable for prediction; and $\boldsymbol{Z} = \{\boldsymbol{Z}_c, \boldsymbol{Z}_s\}$ denotes the latent, high-level multidimensional representations responsible for generating $\boldsymbol{X}$. Our proposed SCM inherits several fundamental settings, including the decomposition of latent representations $\boldsymbol{Z}$ into causal representation $\boldsymbol{Z}_c$ and spurious representation $\boldsymbol{Z}_s$, and the generation of $\boldsymbol{X}$ by both $\boldsymbol{Z}_c$ and $\boldsymbol{Z}_s$. However, some existing approaches assume that spurious correlations arise from either direct causal relations or a latent confounder between $\boldsymbol{Z}_c$ and $\boldsymbol{Z}_s$. Such an assumption may fall short in accounting for various types of biases present in the given data distribution. We incorporate unobserved variables, denoted as $\boldsymbol{U} = \{U_{xy}, U_x\}$, to encode external sources of factors. In particular, $U_{xy}$ represents the latent confounder that impacts both $\boldsymbol{Z}_s$ and $Y$, creating spurious correlations between $\boldsymbol{X}$ and $Y$ through the path $\boldsymbol{X} \leftarrow \boldsymbol{Z}_s \leftarrow U_{xy} \rightarrow Y$. The domain-specific factor $U_x$ affects $\boldsymbol{X}$ via the spurious representation $\boldsymbol{Z}_s$ and influences $Y$ through the path $U_x \rightarrow \boldsymbol{Z}_s \rightarrow \boldsymbol{X} \leftarrow \boldsymbol{Z}_c \leftarrow Y$ given $\boldsymbol{X}$. By incorporating $U_x$ and $U_{xy}$, our SCM explicitly addresses two prevalent data biases: selection bias and stereotype bias, which we will further elaborate on with examples later. Additionally, we select $\boldsymbol{Z}_c$ as the anti-causal features of $Y$ because empirical evidence has demonstrated their superior performance in prediction tasks compared to causal features (Schölkopf et al., 2012; Kilbertus et al., 2018; Lopez-Paz et al., 2017). Compared to Wang & Jordan (2021); Lu et al. (2021), the differences between the SCMs lie in the modeling of high-level latent factors $U_x$ and $U_{xy}$. Mao et al. (2022) and Kong et al. (2022) adopt the same high-level latent factors as ours. However, they assume that causal features as the parent variables to target while we use the child variables.

In accordance with our proposed SCM, traditional deep learning models employed for prediction tasks may falter when confronted with distribution shifts. Under a probabilistic framework, the primary goal of prediction tasks is to estimate the probability $p(Y|\boldsymbol{X})$. We consider two distinct data distributions: the source data distribution $\pi^s$ and the target data distribution $\pi^t$. The training data originates from $\pi^s$, while the testing data is drawn from $\pi^t$. Both of these data distributions are generated within the framework of the same causal structure illustrated in Figure 1. We denote the distributions for the source distribution as $p^s(\cdot)$ and those for the target distribution as $p^t(\cdot)$. The distribution of latent domain-specific factor $U_x$ exhibits variations between $\pi^s$ and $\pi^t$, i.e., $p^s(U_x) \neq p^t(U_x)$. We further assume that the confounding effects remain invariant. This implies

that the distribution of the latent confounder and its causal mechanisms concerning $Z_s$ and $Y$ also remains invariant, i.e., $p^s(U_{xy}) = p^t(U_{xy})$. The probability $p(Y|\boldsymbol{X})$, which is captured by traditional machine learning models, unfortunately takes into account the covariance between $U_x$, $\boldsymbol{X}$, and $Y$ through the path $U_x \to \boldsymbol{Z}_s \to \boldsymbol{X} \leftarrow \boldsymbol{Z}_c \leftarrow Y$ when $\boldsymbol{X}$ is known. Due to the varying distributions of $p(U_x)$, $p(Y|X)$ fluctuates with distribution shifts, i.e., $p^s(Y|\boldsymbol{X}) \neq p^t(Y|\boldsymbol{X})$.

To illustrate our concept, we employ an image from the Waterbird dataset to provide a conceptual understanding of representations $\boldsymbol{Z}_c$ and $\boldsymbol{Z}_s$. For image classification tasks, $\boldsymbol{Z}_c$ typically contains information intrinsic to the object itself, such as its shape or color (Lopez-Paz et al., 2017). In contrast, $\boldsymbol{Z}_s$ extracts information from other aspects of the image, such as the background. For instance, in the case of an image featuring a waterbird flying over the sea, $\boldsymbol{Z}_s$ represents the features responsible for generating the sea in the background. The latent confounder $U_{xy}$ is the high-level factor that naturally leads to the co-occurrence of a water background with the waterbird object. For example, the temperature, altitude, or humidity. We denote the bias that is caused by a high-level invariant confounder between spurious features and the target as the stereotype bias. However, domain-specific factor $U_x$ can introduce spurious co-occurrences between the background and the object. By influencing the distribution of $\boldsymbol{X}$ through data acquisition, $U_x$ can select images in the training set in such a way that the water background co-occurs with the water bird. We refer to these types of biases as selection bias.

These distribution shifts, caused by variations in $U_x$, directly result in changes in $\boldsymbol{Z}_s$. Consequently, the spurious co-occurrences between $\boldsymbol{Z}_s$ and $Y$ vary under distribution shifts and can compromise model accuracy. For instance, a model that leverages the water background for predicting waterbirds may fail when presented with an image of a waterbird against a ground background. Our intention is to encourage the model to rely on invariant features as opposed to features that may exhibit strong but unstable statistical correlations for prediction. Numerous studies have suggested the adoption of an intervention mechanism to eliminate paths involving variables vulnerable to distribution shifts. The primary technical challenge lies in effectively integrating and deploying this interventional mechanism without acquiring information for domain-specific and confounding factors. As a solution, we propose the estimation of an interventional distribution, denoted as $p(Y|\boldsymbol{X}, do(\boldsymbol{Z}_s))$. This distribution describes a scenario where the influence from $U_x$ and $U_{xy}$ to $\boldsymbol{X}$ is mitigated through the intervention on the representation $\boldsymbol{Z}_s$. Hence, $p(Y|\boldsymbol{X}, do(\boldsymbol{Z}_s))$ effectively prevents the $U_x$ from influencing $Y$, and is invariant and transportable across domains, i.e., $p^s(Y|\boldsymbol{X}, do(\boldsymbol{Z}_s)) = p^t(Y|\boldsymbol{X}, do(\boldsymbol{Z}_s))$. We then identify and estimate the $p(Y|\boldsymbol{X}, do(\boldsymbol{Z}_s))$ in accordance with the proposed SCM, utilizing the corresponding observational distributions. This process is detailed in Section 3.2.

## 3.2 CAUSAL INTERVENTIONAL INFERENCE

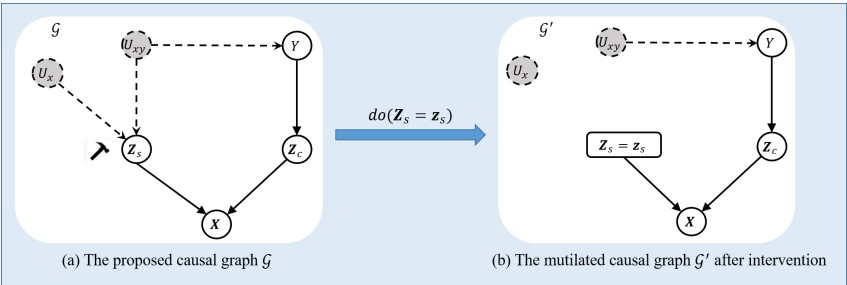

(a) The proposed causal graph $\mathcal{G}$      (b) The mutilated causal graph $\mathcal{G}'$ after intervention

Figure 2: Intervention on the proposed SCM, where (a) is the original graph and (b) is the mutilated graph after intervention on $\boldsymbol{Z}_s$.

Intuitively, we aim to infer from the interventional regime, where all the arrows into variable $\boldsymbol{Z}_s$ are removed as shown in Figure 2(b). We denote the original graph before intervention in Figure. 2(a) as $\mathcal{G}$ and the interventional graph in Figure. 2(b) as $\mathcal{G}'$ (as known as the mutilated graph). The difference between $\mathcal{G}$ and $\mathcal{G}'$ is the removal of links $\boldsymbol{U}_{xy} \to \boldsymbol{Z}_s$ and $\boldsymbol{U}_x \to \boldsymbol{Z}_s$. A key contribution of this paper is the ability to represent the distribution of $y$ given $x$ in the mutilated graph as a function of observational distributions in the original graph. **Theorem 3.1** demonstrates that the interventional distribution $p(y|\boldsymbol{x}, do(\boldsymbol{z}_s))$ can be computed using a set of observational distributions.

**Theorem 3.1.** *Subject to the SCM in Figure 2(a), the interventional distribution $p(y|\boldsymbol{x}, do(\boldsymbol{z}_s))$ can be computed through Eq. (1) with a set of observational distributions.*

$$p(y|\boldsymbol{x}, do(\boldsymbol{z}_s)) = \frac{\int_{\boldsymbol{z}_c} p(y|\boldsymbol{z}_c)p(\boldsymbol{x}|\boldsymbol{z}_c, \boldsymbol{z}_s)p(\boldsymbol{z}_c)\,\mathrm{d}\boldsymbol{z}_c}{\int_{\boldsymbol{z}_c} p(\boldsymbol{x}|\boldsymbol{z}_c, \boldsymbol{z}_s)p(\boldsymbol{z}_c)\,\mathrm{d}\boldsymbol{z}_c} \tag{1}$$

We provide detailed proof in Appendix A. $p(y|\boldsymbol{x}, do(\boldsymbol{z}_s))$ is invariant across domains **when given** $\boldsymbol{x}$ **and** $\boldsymbol{z}_s$. However, it is important to note that $p(y|\boldsymbol{x}, do(\boldsymbol{z}_s))$ depends on $\boldsymbol{z}_s$. To infer the label for an input $\boldsymbol{x}^t$, it is imperative to utilize the true value of $\boldsymbol{Z}_s$ for $\boldsymbol{x}^t$ to accurately account for the confounding effects. We construe $p(\boldsymbol{z}_s|\boldsymbol{x})$ as a proxy distribution that enables us to derive the true values of $\boldsymbol{Z}_s$ from an input $\boldsymbol{X}$. We will elaborate in Section 4.2 about how to approximate this proxy distribution. By applying **Theorem 3.1**, we have

$$\mathbb{E}_{p(\boldsymbol{z}_s|\boldsymbol{x})}[p(y|\boldsymbol{x}, do(\boldsymbol{z}_s))] \approx \frac{1}{N}\sum_{n=1}^{N}[\sum_{l=1}^{L} p(y|\boldsymbol{z}_{c,l})\omega(\boldsymbol{z}_{c,l}, \boldsymbol{z}_{s,n})], \omega(\boldsymbol{z}_{c,l}, \boldsymbol{z}_{s,n}) = \frac{p(\boldsymbol{x}|\boldsymbol{z}_{c,l}, \boldsymbol{z}_{s,n})}{\sum_{l'=1}^{L} p(\boldsymbol{x}|\boldsymbol{z}_{c,l'}, \boldsymbol{z}_{s,n})} \tag{2}$$

where $\boldsymbol{z}_{c,l} \sim p(\boldsymbol{z}_c), \boldsymbol{z}_{s,n} \sim p(\boldsymbol{z}_s|\boldsymbol{x})$. The intuition of the interventional distribution $\mathbb{E}_{p(\boldsymbol{z}_s|\boldsymbol{x})}[p(y|\boldsymbol{x}, do(\boldsymbol{z}_s))]$ is that instead of performing inference using a single $\boldsymbol{z}_c$ through $p(y|\boldsymbol{z}_c)$, we utilize a set of $\boldsymbol{z}_{c,l}$ through the linear combination of $p(y|\boldsymbol{z}_{c,l})$ weighted by $\boldsymbol{w}(\boldsymbol{z}_{c,l}, \boldsymbol{z}_{s,n})$. The weight $\boldsymbol{w}(\boldsymbol{z}_{c,l}, \boldsymbol{z}_{s,n})$ value is high for samples whose $\boldsymbol{z}_c$ is compatible with that of $\boldsymbol{x}$.

According to Eq. (2), calculating $\mathbb{E}_{p(\boldsymbol{z}_s|\boldsymbol{x})}[p(y|\boldsymbol{x}, do(\boldsymbol{z}_s))]$ necessitates observational distributions such as $p(y|\boldsymbol{z}_c), p(\boldsymbol{x}|\boldsymbol{z}_c, \boldsymbol{z}_s), p(\boldsymbol{z}_c)$, as well as the distributions $p(\boldsymbol{z}_c|\boldsymbol{x}), p(\boldsymbol{z}_s|\boldsymbol{x})$ to extract $\boldsymbol{z}_c, \boldsymbol{z}_s$ from the input $\boldsymbol{x}$. Since we can approximate $p(\boldsymbol{z}_c) = \int p(\boldsymbol{z}_c|\boldsymbol{x})p(\boldsymbol{x})d\boldsymbol{x}$ by marginalizing $\boldsymbol{x}$, our goal is to develop a causal representation learning framework that constructs and estimates the distributions $p(y|\boldsymbol{z}_c), p(\boldsymbol{x}|\boldsymbol{z}_c, \boldsymbol{z}_s), p(\boldsymbol{z}_c|\boldsymbol{x})$, and $p(\boldsymbol{z}_s|\boldsymbol{x})$ based on the proposed SCM, all from a single observational dataset. Subsequently, we utilize these distributions to construct the interventional distribution for inferring labels for testing data.

## 4 CAUSAL REPRESENTATION LEARNING AND INTERVENTIONAL INFERENCE

We propose a framework for performing causal representation learning and interventional inference. The framework comprises three steps. Given the SCM architecture, we first parameterize the SCM with conditional distributions. We then model the conditional distributions using neural networks and learn their parameters by minimizing the logarithm marginal likelihood over observed variables, denoted as $-\log p(\boldsymbol{x}, y)$. With the estimated distributions, we are able to calculate interventional distributions for inference. We introduce the procedures of SCM parameterization and learning in Section 4.1, while introducing the inference procedure in Section 4.2.

### 4.1 SCM-GUIDED REPRESENTATION LEARNING

**SCM parameterization:** To parameterize the proposed SCM, we factorize the joint distribution of all the variables using the chain rule, as illustrated in Eq. (3).

$$p(\boldsymbol{x}, y, \boldsymbol{z}_c, \boldsymbol{z}_s, u_x, u_{xy}) = p(u_x)p(u_{xy})p(y|u_{xy})p(\boldsymbol{z}_c|y)p(\boldsymbol{z}_s|u_x, u_{xy})p(\boldsymbol{x}|\boldsymbol{z}_c, \boldsymbol{z}_s) \tag{3}$$

Notably, we employ Bayes' theorem and transform $p(\boldsymbol{z}_c|y)$ into $\frac{p(y|\boldsymbol{z}_c)p(\boldsymbol{z}_c)}{p(y)}$ to explicitly model $p(y|\boldsymbol{z}_c)$ for classification. Since $z_c, z_s$ are unobserved, we leverage expressive neural networks to parameterize the corresponding conditional distributions. Especially, we model $p(\boldsymbol{x}|\boldsymbol{z}_c, \boldsymbol{z}_s)$ with a decoder and parameter $\Phi$, $p(y|\boldsymbol{z}_c)$ with a classifier and parameter $\Psi$. We regard the distributions related to $u_x$ and $u_{xy}$ as prior distributions. To learn the unobserved representation $\boldsymbol{z}_c$ and $\boldsymbol{z}_s$ given $\boldsymbol{x}$, we introduce a variational distribution $q(\boldsymbol{z}_c, \boldsymbol{z}_s|\boldsymbol{x})$ as approximation for $p(\boldsymbol{z}_c, \boldsymbol{z}_s|\boldsymbol{x})$. We model $q(\boldsymbol{z}_c, \boldsymbol{z}_s|\boldsymbol{x})$ with encoders and parameter $\Theta$. We further assume that $q(\boldsymbol{z}_c, \boldsymbol{z}_s|\boldsymbol{x}) = q(\boldsymbol{z}_c|\boldsymbol{x})q(\boldsymbol{z}_s|\boldsymbol{x})$ and parameterizes two encoders $q(\boldsymbol{z}_c|\boldsymbol{x}), q(\boldsymbol{z}_s|\boldsymbol{x})$ with $\Theta_c, \Theta_s$ respectively.

**SCM parameters learning:** Subject to the above parameterization, we learn the parameters $\{\Theta, \Phi, \Psi\}$ by minimizing an upper bound of $-\log p(\boldsymbol{x}, y)$, as outlined in Eq. (4):

$$-\log p(\boldsymbol{x}, y) \leq \mathcal{L}_{obj}(\boldsymbol{x}, y, \Theta, \Phi, \Psi)$$
$$= -\mathbb{E}_{p(u_x, u_{xy})}KL\big(q_{\Theta}(\boldsymbol{z}_s|\boldsymbol{x})||p(\boldsymbol{z}_s|u_x, u_{xy})\big) - KL\big(q_{\Theta_c}(\boldsymbol{z}_c|\boldsymbol{x})||p(\boldsymbol{z}_c)\big) \tag{4}$$
$$+ \mathbb{E}_{q_{\Theta_c}(\boldsymbol{z}_c|\boldsymbol{x})}[\log p_{\Psi}(y|\boldsymbol{z}_c)] + \mathbb{E}_{q_{\Theta}(\boldsymbol{z}_c, \boldsymbol{z}_s|\boldsymbol{x})}[\log p_{\Phi}(\boldsymbol{x}|\boldsymbol{z}_c, \boldsymbol{z}_s)]$$

The detailed derivations can be found in Appendix B.2. Guided by the training objective, we construct a learning framework with two encoders, a decoder, and a classifier, as illustrated in Figure 3.

The two KL divergences serve as regularization, aligning the encoder distributions with prior distributions, notably $p(\boldsymbol{z}_s|u_x, u_{xy})$ and $p(\boldsymbol{z}_c)$. Essentially, the spurious representation $\boldsymbol{z}_s$ varies with changes in $u_x$, reflecting their domain-specific and latent confounding effects. $U_x$ denotes any information specific to the domain. We adopt a common simplification procedure to assume that $U_x$ is a discrete variable represents the domain index (Lu et al., 2021). $p(U_x)$ follows a categorical distribution. Additionally, we further assume $q(\boldsymbol{z}_s|\boldsymbol{x})$, $q(\boldsymbol{z}_c|\boldsymbol{x})$, and $p(\boldsymbol{z}_s|u_x, u_{xy})$ follow Gaussian distributions. $p(\boldsymbol{z}_s|u_x, u_{xy})$ has distinct means and variances with different values of $u_x$. We collectively refer

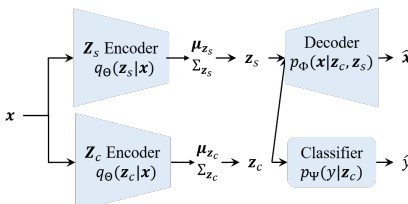

Figure 3: The proposed causal representation learning framework

to these variables as $U = \{U_x, U_{xy}\}$ and the distinct classes of $U$ are associated with the diverse values of $U_x$.. In practice, we need to pre-determine the number of classes for the variable $U$. After each epoch, we cluster the learned $\boldsymbol{z}_s$ values and adjust the means and variances for $p(\boldsymbol{z}_s|u_x, u_{xy})$ based on empirical values from each cluster. Without extra data, we assume $p(\boldsymbol{z}_c)$ adheres to a standard multivariate normal distribution. We summarize the training procedure in Algorithm 1.

---

**Algorithm 1** Causal Representation Learning Procedure

---

1: **Input:** Training set $\mathcal{D}$ over $\{(\boldsymbol{X}, Y)\}$; $p(U)$; $|U| = J$.
2: **Goal:** Estimate $q_{\Theta_c}(\boldsymbol{Z}_c|\boldsymbol{X})$, $q_{\Theta_s}(\boldsymbol{Z}_s|\boldsymbol{X})$, $p_\Phi(\boldsymbol{X}|\boldsymbol{Z}_s, \boldsymbol{Z}_c)$, and $p_\Psi(Y|\boldsymbol{Z}_c)$.
3: Initialize encoders, decoder, and classifier.
4: **for** $i = 1, 2, \cdots, M$ **do**
5:     Obtain one input observation $\boldsymbol{x}^i$
6:     Input $\boldsymbol{X} = \boldsymbol{x}^i$ into the encoder $q_{\Theta_s}(\boldsymbol{Z}_s|\boldsymbol{X})$, obtain $\boldsymbol{z}_s^i = \boldsymbol{\mu}_{\boldsymbol{z}_s}(\boldsymbol{x}^i)$
7: **end for**
8: Cluster $\{\boldsymbol{z}_s^i\}_{i=1}^M$ into $J$ bins with Kmeans algorithm. For each bin $U = u_j$, we set the mean and variance for $p(\boldsymbol{Z}_s|U = u_j)$ as the empirical mean and variance of the $\boldsymbol{z}_s$s in this bin.
9: **repeat**
10:     **for** $i = 1, 2, \cdots, M$ **do**
11:         Obtain one input observation and its label $(\boldsymbol{x}^i, y^i)$ from training batch.
12:         Sample $\boldsymbol{z}_s^i \sim q_{\Theta_s}(\boldsymbol{z}_s|\boldsymbol{x}^i)$, $\boldsymbol{z}_c^i \sim q_{\Theta_c}(\boldsymbol{z}_c|\boldsymbol{x}^i)$.
13:         Input $\boldsymbol{Z}_s = \boldsymbol{z}_s^i$, $\boldsymbol{Z}_c = \boldsymbol{z}_c^i$ into the decoder and compute $p_\Phi(\boldsymbol{x}^i|\boldsymbol{z}_c^i, \boldsymbol{z}_s^i)$.
14:         Input $\boldsymbol{Z}_c = \boldsymbol{z}_c^i$ into the classifier and compute $p_\Psi(y^i|\boldsymbol{z}_c^i)$.
15:     **end for**
16:     Update $\Theta, \Phi, \Psi$ by minimizing the training objective in Eq. (4) via gradient descent.
17:     **for** $i = 1, 2, \cdots, M$ **do**
18:         Obtain one input observation $\boldsymbol{x}^i$
19:         Input $\boldsymbol{X} = \boldsymbol{x}^i$ into the encoder $q_{\Theta_s}(\boldsymbol{Z}_s|\boldsymbol{X})$, obtain $\boldsymbol{z}_s^i = \boldsymbol{\mu}_{\boldsymbol{z}_s}(\boldsymbol{x}^i)$
20:     **end for**
21:     Cluster $\{\boldsymbol{z}_s^i\}_{i=1}^M$ into $J$ bins with Kmeans algorithm. For each bin $U = u_j$, we update the mean and variance for $p(\boldsymbol{Z}_s|U = u_j)$ as the empirical mean and variance of the $\boldsymbol{z}_s$s in this bin.
22: **until** Converge
23: **Output** $q_{\hat{\Theta}_c}(\boldsymbol{Z}_c|\boldsymbol{X})$, $q_{\hat{\Theta}_s}(\boldsymbol{Z}_s|\boldsymbol{X})$, $p_{\hat{\Phi}}(\boldsymbol{X}|\boldsymbol{Z}_c, \boldsymbol{Z}_s)$, and $p_{\hat{\Psi}}(Y|\boldsymbol{Z}_c)$.

---

**Intuition of the learning framework:** Our causal representation learning captures intrinsic mechanisms from a provided causal graph using partially observed data. Unlike conventional parameter learning methods that handle missing data, our framework employs deep learning models to tackle the intricate mapping from the input $\boldsymbol{x}$ to the representations $\boldsymbol{z}_s$ and $\boldsymbol{z}_c$. The primary challenge in our learning procedure stems from the presence of a substantial number of latent variables. Among the six variables of interest in the SCM, we only have access to the values of $\boldsymbol{X}$ and $Y$. The absence of observations for $\boldsymbol{Z}_s$, $U_x$ and $U_{xy}$ necessitates our framework to rely on strong assumptions. We also need to estimate $p(\boldsymbol{z}_s|u_x, u_{xy})$ during training in order to effectively regularize the learning of $q(\boldsymbol{z}_s|\boldsymbol{x})$. Nevertheless, despite these strong assumptions, our framework demonstrates the capability to identify and learn causal representations, even when operating solely within a single observational training domain and without requiring any domain/index knowledge.

**Identifiability of representation $\boldsymbol{Z} = \{\boldsymbol{Z}_c, \boldsymbol{Z}_s\}$:** Establishing the identifiability of the representations $\boldsymbol{Z}$ is a crucial prerequisite for enabling causal intervention within our framework. We assess

the identifiability of the learned $\boldsymbol{Z}$ in our model, leveraging advanced theoretical results in the field.

**Proposition 4.1.** *If the data distribution is generated via the SCM in Figure 1, the obtained $\boldsymbol{Z}$ from within our training framework is identifiable up to an affine transformation.*

Kivva et al. (2022) provides rigorous theoretical guarantees regarding the identifiability of the learned $\boldsymbol{Z}$ subject to a set of assumptions governing $p(\boldsymbol{z})$ and mapping function from $\boldsymbol{z}$ to $\boldsymbol{x}$, denoted as $f$. These assumptions encompass the following aspects: (1) $p(\boldsymbol{z})$ follows a distribution with the format $p(\boldsymbol{z}) = \sum_{j=1}^{J} \lambda_j \mathcal{N}(\mu_j, \Sigma_j), J \geq 1$; (2) $f$ takes the form of a piecewise affine function; (3) $f$ is injective. In particular, our formulation of $p(\boldsymbol{z}_s)$ resembles a mixture Gaussian distribution: $p(\boldsymbol{z}_s) = \sum_{j=1}^{J} \lambda_j \mathcal{N}(\mu_j, \Sigma_j)$ with $p(U = u_j) = \lambda_j$ and $|U| = J$. Meanwhile, $p(\boldsymbol{z}_c)$ adheres to a standard normal distribution, satisfies the assumption (1) by setting $J = 1$. Furthermore, a multi-layer perceptron utilizing leaky ReLU activations can be recognized as a piecewise affine function. Based on Corollary H.4 in Kivva et al. (2022), if such a function uses leaky ReLU activations and features a monotonically increasing number of hidden neurons, it's considered injective. Therefore, in our framework, we design $f$ (serving as the decoder) as leaky ReLU networks that match the encoders in complexity and fulfill assumptions (2) and (3). This identifiability assurance empowers us to discern $\boldsymbol{Z}_c, \boldsymbol{Z}_s$. This, in turn, allows us to declare our ability to intervene on the desired representation and obtain the invariant interventional distribution to infer from.

**Disentanglement between $\boldsymbol{z}_s$ and $\boldsymbol{z}_c$:** Our learning process disentangles $\boldsymbol{z}_s$ from $\boldsymbol{z}_c$ using asymmetric regularizations such as the distinct prior assumptions for $p(\boldsymbol{z}_s)$ and $p(\boldsymbol{z}_c)$ and the inclusion of downstream tasks for $\boldsymbol{z}_c$. It's worth noting that our framework, which does not require domain-specific knowledge, cannot theoretically guarantee a perfect disentanglement between these representations. This limitation serves as a motivation for our method to perform inference from interventional distribution. If the anti-causal representation $\boldsymbol{z}_c$ were perfectly disentangled, one could obtain theoretically optimal OOD prediction by inferring directly from $p(y|\boldsymbol{z}_c)$. However, in cases where $\boldsymbol{z}_c$ contains some degree of $\boldsymbol{z}_s$ information, interventional inference, which estimates a weighted expectation of $p(y|\boldsymbol{z}_c)$, can effectively mitigate errors arising from the representation learning process and outperform the OOD prediction performance compared to inferring solely from $p(y|\boldsymbol{z}_c)$. Empirical results on multiple benchmark distribution shift datasets have confirmed these assertions.

## 4.2 INTERVENTIONAL INFERENCE

The framework for learning causal representation furnishes the observational distributions required to compute interventional distributions. For performing interventional inference of an input $\boldsymbol{x}^t$, our goal is to infer its label from the expected interventional distribution $\mathbb{E}_{p(\boldsymbol{z}_s|\boldsymbol{x}^t)}[p(y|\boldsymbol{x}^t, do(\boldsymbol{z}_s))]$. In particular, we approximate $p(\boldsymbol{z}_s|\boldsymbol{x})$ with $q_{\hat{\Theta}_s}(\boldsymbol{z}_s|\boldsymbol{x})$, $p(\boldsymbol{z}_c|\boldsymbol{x})$ with $q_{\hat{\Theta}_c}(\boldsymbol{z}_c|\boldsymbol{x})$[1]. We observe from Figure 1 that $\boldsymbol{X}$ is independent of $U$ given $\boldsymbol{Z} = [\boldsymbol{Z}_c, \boldsymbol{Z}_s]$, rendering $p(\boldsymbol{X}|\boldsymbol{Z})$ invariant. Therefore, we choose to obtain the identifiable $\boldsymbol{z}_s^t$ from $q_{\hat{\Theta}}(\boldsymbol{z}|\boldsymbol{x}^t)$ with a high $p(\boldsymbol{x}^t|\boldsymbol{z}^t)$. However, as $q(\boldsymbol{z}_s|\boldsymbol{x}^t)$ represents a variational estimation of the desired proxy distribution $p(\boldsymbol{z}_s|\boldsymbol{x}^t)$, we average the interventional distribution over multiple samples of $\boldsymbol{z}_s$ from $q(\boldsymbol{z}_s|\boldsymbol{x}^t)$. In practice, randomly obtaining a $\boldsymbol{z}_s$ value is likely to yield a low $p(\boldsymbol{X}|\boldsymbol{Z})$ and contribute minimally to the calculation of the interventional distribution. However, due to the high dimensionality of the input data $\boldsymbol{x}$, the differences between $p(\boldsymbol{x}|\boldsymbol{z}_{c,l}, \boldsymbol{z}_{s,n})$s with distinct $\boldsymbol{z}_{c,l}$ are significantly large. This results in a weight $\boldsymbol{\omega}(\boldsymbol{z}_{c,l}, \boldsymbol{z}_{s,n})$ that is close to 1 for the $\boldsymbol{z}_{c,l}$ with the highest probability. This situation contradicts our objective of mitigating the spurious correlations' influence through summation and could lead to erroneous predictions due to the imperfect disentanglement of $\boldsymbol{z}_c$ and $\boldsymbol{z}_s$. Hence, we employ Baye's theorem to transform $p(\boldsymbol{x}|\boldsymbol{z}_c, \boldsymbol{z}_s)$ into $p(\boldsymbol{z}_c, \boldsymbol{z}_s|\boldsymbol{x})$, which has lower dimensions.

$$\boldsymbol{\omega}(\boldsymbol{z}_{c,l}, \boldsymbol{z}_{s,n}) = \frac{p(\boldsymbol{x}|\boldsymbol{z}_{c,l}, \boldsymbol{z}_{s,n})}{\sum_{l'=1}^{L} p(\boldsymbol{x}|\boldsymbol{z}_{c,l'}, \boldsymbol{z}_{s,n})} = \frac{\frac{p(\boldsymbol{z}_{c,l}, \boldsymbol{z}_{s,n}|\boldsymbol{x})}{p(\boldsymbol{z}_{c,l}, \boldsymbol{z}_{s,n})}}{\sum_{l'=1}^{L} \frac{p(\boldsymbol{z}_{c,l'}, \boldsymbol{z}_{s,n}|\boldsymbol{x})}{p(\boldsymbol{z}_{c,l'}, \boldsymbol{z}_{s,n})}} \tag{5}$$

We can estimate $p(\boldsymbol{z}_c, \boldsymbol{z}_s)$ using $p(\boldsymbol{z}_c, \boldsymbol{z}_s|\boldsymbol{x})$ as follows: $p(\boldsymbol{z}_c, \boldsymbol{z}_s) = \int_{\boldsymbol{x}} p(\boldsymbol{z}_c, \boldsymbol{z}_s|\boldsymbol{x}) p(\boldsymbol{x}) \, d\boldsymbol{x} \approx \frac{1}{L} \sum_{k=1}^{L} p(\boldsymbol{z}_c, \boldsymbol{z}_s|\boldsymbol{x}^k), \boldsymbol{x}^k \sim p(\boldsymbol{x})$. Given these approximations, we can compute the expected

---

[1]Since we approximate $p(\boldsymbol{z}_c, \boldsymbol{z}_s|\boldsymbol{x})$ using $q(\boldsymbol{z}_c, \boldsymbol{z}_s|\boldsymbol{x})$ and further assume $q(\boldsymbol{z}_c, \boldsymbol{z}_s|\boldsymbol{x}) = q(\boldsymbol{z}_c|\boldsymbol{x})q(\boldsymbol{z}_s|\boldsymbol{x})$, $p(\boldsymbol{z}_c|\boldsymbol{x})$ and $p(\boldsymbol{z}_s|\boldsymbol{x})$ can be approximated by $q(\boldsymbol{z}_c|\boldsymbol{x})$ and $q(\boldsymbol{z}_s|\boldsymbol{x})$ respectively. For example, $p(\boldsymbol{z}_s|\boldsymbol{x}) = \int_{\boldsymbol{z}_c} p(\boldsymbol{z}_s, \boldsymbol{z}_c|\boldsymbol{x}) \, d\boldsymbol{z}_c \approx \int_{\boldsymbol{z}_c} q(\boldsymbol{z}_s, \boldsymbol{z}_c|\boldsymbol{x}) \, d\boldsymbol{z}_c = \int_{\boldsymbol{z}_c} q(\boldsymbol{z}_s|\boldsymbol{x})q(\boldsymbol{z}_c|\boldsymbol{x}) \, d\boldsymbol{z}_c = q(\boldsymbol{z}_s|\boldsymbol{x})$

interventional distribution using only $p_{\hat{\Psi}}(Y|\boldsymbol{Z}_c)$ and $q_{\hat{\Theta}}(\boldsymbol{Z}_c, \boldsymbol{Z}_s|\boldsymbol{X})$, as outlined in Equation (6).

$$\mathbb{E}_{p(\boldsymbol{z}_s|\boldsymbol{x}^t)}[p(y|\boldsymbol{x}^t, do(\boldsymbol{z}_s))] = \frac{1}{N}\sum_{n=1}^{N}[\sum_{l=1}^{L} p_{\hat{\Psi}}(y|\boldsymbol{z}_{c,l})\boldsymbol{\omega}(\boldsymbol{z}_{c,l}, \boldsymbol{z}_{s,n}^t)]$$

$$\boldsymbol{\omega}(\boldsymbol{z}_{c,l}, \boldsymbol{z}_{s,n}^t) = \frac{\frac{q_{\hat{\Theta}}(\boldsymbol{z}_{c,l}, \boldsymbol{z}_{s,n}^t|\boldsymbol{x}^t)}{\sum_{k=1}^{L} q_{\hat{\Theta}}(\boldsymbol{z}_{c,l}, \boldsymbol{z}_{s,n}^t|\boldsymbol{x}^k)}}{\sum_{l'=1}^{L} \frac{q_{\hat{\Theta}}(\boldsymbol{z}_{c,l'}, \boldsymbol{z}_{s,n}^t|\boldsymbol{x}^t)}{\sum_{k=1}^{L} q_{\hat{\Theta}}(\boldsymbol{z}_{c,l'}, \boldsymbol{z}_{s,n}^t|\boldsymbol{x}^k)}} \quad \boldsymbol{z}_{s,n}^t \sim q_{\hat{\Theta}_s}(\boldsymbol{z}_s|\boldsymbol{x}^t), \boldsymbol{z}_{c,l} \sim q_{\hat{\Theta}_c}(\boldsymbol{z}_c|\boldsymbol{x}^l), \boldsymbol{x}^l, \boldsymbol{x}^k \sim \mathcal{D}$$

(6)

The procedure for interventional inference is detailed in Algorithm 2 in Appendix C. For efficiency, we do not enumerate $\boldsymbol{z}_c$ for every training input. We choose $L$ samples of $\boldsymbol{z}_{c,l}$ based on the top weights $\boldsymbol{w}(\boldsymbol{z}_{c,l}, \boldsymbol{z}_{s,n}^t)$. The optimal $L$ value is determined empirically during inference.

## 5 EXPERIMENTS

We demonstrate the effectiveness of our proposed method, which we denote as **C**ausal **R**epresentation **L**earning and **I**nterventional **I**nference (CRLII), in terms of OOD prediction. We conduct experiments on both the **synthetic and real** benchmark distribution shift datasets against state-of-the-art domain generalization (DG) baselines.

**Datasets.** We evaluate our CRLII method on a synthetic dataset, Colored-MNIST, and three real datasets: PACS, VLCS, and OfficeHome. **CMNIST** (Mao et al., 2022) contains digit images from 10 different categories.[2] In the training domain, the data is generated such that digits are associated with different background or foreground colors. However, in the testing domain, the digits' colors are independent. **PACS** (Li et al., 2017) contains images from four domains: Photo (P), Art painting (A), Cartoon (C), and Sketch (S), with each domain comprising images in 7

Table 1: Comparison with SOTA methods on CMNIST data set.

| Algorithms | Prediction Acc (%) | |
|---|---|---|
| | In-distribution | OOD |
| ERM | **99.6** | 12.3 |
| RSC | 96.3 | 20.5 |
| IRM | 98.4 | 19.9 |
| GenInt | 58.5 | 31.6 |
| CTrans | 82.9 | 51.4 |
| CRLII | 96.0 | **69.8** |

categories. **VLCS** (Torralba & Efros, 2011) has images of 5 categories from four domains: PASCAL VOC 2007 (P), LabelMe (L), Caltech (C), and Sun (S). **OfficeHome** (Venkateswara et al., 2017) includes images from four domains: Artistic (A), Clipart (C), Product (P), and Real World (R), with 65 object categories related to office and home settings. We use the standard leave-one-domain-out protocol, as per previous domain generalization methods, testing on images from one domain and training on the others.

**Baselines.** We compare with three types of approaches. First, we compare to the correlation-based classifier and adopt the state-of-the-art classifier as the ERM approach for each dataset. We then compare with causal DG approaches, including IRM(Arjovsky et al., 2019), GenInt(Mao et al., 2021), CTrans(Mao et al., 2022), SageNet(Nam et al., 2021), MatchDG(Mahajan et al., 2021). For a comprehensive comparison, especially on real datasets, we also compare to other DG methods, such as DRO(Sagawa et al., 2019), MLDG(Li et al., 2017), CORAL(Sun & Saenko, 2016), RSC(Huang et al., 2020), Mixup(Yan et al., 2020), etc.

**Implementation Details.** For CMNIST, we use a two-layer MLP for our encoders and decoder. For PACS, VLCS, and Office-Home, we utilize a pre-trained ResNet-50 on ImageNet as the encoder backbone and select a decoder of comparable complexity. We set the number of classes for $|U|$ to 2 for CMNIST and 3 for the other datasets. Results are averaged over 5 trials. For detailed hyperparameter choices and ablation studies, see the Appendix.

We present the empirical results for the synthetic dataset in Table 1, and for the real datasets in Tables 2 and 3. As evident from Table 1, our CRLII method achieves optimal OOD performance on the CMNIST dataset while ensuring comparable in-distribution accuracy. Notably, CRLII significantly outperforms leading methods such as CTrans, GenInt, and RSC by a margin of at least 18.4%. In our experiment setup, each combination

---

[2]We adopt the most challenging setting of the colored MNIST dataset as one of our baselines (Mao et al., 2022). This setting creates a significant difference between the distributions of training domain images and testing domain images, making correlation-based models capture spurious correlations between color and digits.

of color and digit is treated as a unique scenario for the latent variables $U$ (or a domain). Within this context, $z_s$ captures color information and $z_c$ embodies shape details.

This configuration results in a training dataset spanning two domains, i.e., $|U| = 2$. Empirically, the color information across these domains is distinct, as evidenced by the considerable distance between $p(z_s|u = 0)$ and $p(z_s|u = 1)$. For instance, images of the digit 1 might appear against either a black or cyan background. The significant difference in the distributions of the spurious representation $z_s$ enhances CRLII's ability to effectively disentangle $z_s$ from $z_c$, enabling more precise $z_c$ extraction for interventional inference. Thus, we contend that our method excels on datasets aligning with our SCM assumptions.

Table 2: Comparison with SOTA methods on PACS.

| Algorithms | PACS | | | | |
|---|---|---|---|---|---|
| | **A** | **C** | **P** | **S** | **Avg** |
| ERM | $84.8_{\pm1.3}$ | $76.4_{\pm1.1}$ | $96.7_{\pm0.6}$ | $76.1_{\pm1.0}$ | 83.5 |
| GroupDRO | $83.5_{\pm0.9}$ | $79.1_{\pm0.6}$ | $96.7_{\pm0.3}$ | $78.3_{\pm2.0}$ | 84.4 |
| MLDG | $85.5_{\pm1.4}$ | $80.1_{\pm1.7}$ | $97.4_{\pm0.3}$ | $76.6_{\pm1.1}$ | 84.9 |
| CORAL | $88.3_{\pm0.2}$ | $80.0_{\pm0.7}$ | $97.5_{\pm0.3}$ | $78.8_{\pm1.3}$ | 86.2 |
| MMD | $86.1_{\pm1.4}$ | $79.4_{\pm0.9}$ | $96.6_{\pm0.2}$ | $76.5_{\pm0.7}$ | 84.6 |
| RSC | $85.4_{\pm0.8}$ | $79.7_{\pm1.8}$ | $97.6_{\pm0.3}$ | $78.2_{\pm1.2}$ | 85.2 |
| Mixup | $86.1_{\pm0.7}$ | $78.9_{\pm0.8}$ | $97.6_{\pm0.1}$ | $75.8_{\pm1.8}$ | 84.6 |
| DANN | $86.4_{\pm0.8}$ | $77.4_{\pm0.8}$ | $97.3_{\pm0.4}$ | $73.5_{\pm2.3}$ | 83.6 |
| CDANN | $84.6_{\pm1.8}$ | $75.5_{\pm0.9}$ | $96.8_{\pm0.3}$ | $73.5_{\pm0.6}$ | 82.6 |
| MTL | $87.5_{\pm0.8}$ | $77.1_{\pm0.7}$ | $96.4_{\pm0.8}$ | $77.3_{\pm1.8}$ | 84.6 |
| ARM | $86.8_{\pm0.6}$ | $76.8_{\pm0.7}$ | $97.4_{\pm0.3}$ | $79.3_{\pm1.2}$ | 85.1 |
| IRM | $84.7_{\pm0.4}$ | $80.0_{\pm0.6}$ | $97.2_{\pm0.3}$ | $79.3_{\pm1.0}$ | 85.5 |
| SagNet | $87.4_{\pm1.0}$ | $80.7_{\pm0.6}$ | $97.1_{\pm0.1}$ | $80.0_{\pm0.4}$ | 86.3 |
| MatchDG | $85.7_{\pm1.6}$ | $82.5_{\pm0.7}$ | $\textbf{97.9}_{\pm0.7}$ | $77.3_{\pm1.1}$ | 85.9 |
| CRLII | $\textbf{89.2}_{\pm0.7}$ | $\textbf{84.6}_{\pm1.2}$ | $97.1_{\pm0.5}$ | $\textbf{83.6}_{\pm0.6}$ | **88.6** |

Table 3: Comparison with SOTA methods on VLCS and OfficeHome datasets.

| Algorithms | VLCS | | | | | OfficeHome | | | | |
|---|---|---|---|---|---|---|---|---|---|---|
| | **C** | **L** | **S** | **V** | **Avg** | **A** | **C** | **P** | **R** | **Avg** |
| ERM | $98.0_{\pm0.4}$ | $62.6_{\pm0.9}$ | $70.8_{\pm1.9}$ | $77.5_{\pm1.9}$ | 77.2 | $61.3_{\pm0.7}$ | $52.4_{\pm0.3}$ | $75.8_{\pm0.1}$ | $76.6_{\pm0.3}$ | 66.5 |
| GroupDRO | $98.1_{\pm0.3}$ | $66.4_{\pm0.9}$ | $71.0_{\pm0.3}$ | $76.1_{\pm1.4}$ | 77.9 | $60.4_{\pm0.7}$ | $52.7_{\pm1.0}$ | $75.0_{\pm0.7}$ | $76.0_{\pm0.7}$ | 66.0 |
| MLDG | $98.5_{\pm0.3}$ | $61.7_{\pm1.2}$ | $\textbf{73.6}_{\pm1.8}$ | $75.0_{\pm0.8}$ | 77.2 | $61.5_{\pm0.9}$ | $53.2_{\pm0.6}$ | $75.0_{\pm1.2}$ | $77.5_{\pm0.4}$ | 66.8 |
| CORAL | $96.9_{\pm0.9}$ | $65.7_{\pm1.2}$ | $73.3_{\pm0.7}$ | $78.7_{\pm0.8}$ | 78.7 | $65.3_{\pm0.4}$ | $54.4_{\pm0.5}$ | $76.5_{\pm0.1}$ | $78.4_{\pm0.5}$ | 68.7 |
| MMD | $98.3_{\pm0.1}$ | $65.6_{\pm0.7}$ | $69.7_{\pm1.0}$ | $75.7_{\pm0.9}$ | 77.3 | $60.4_{\pm0.2}$ | $53.3_{\pm0.3}$ | $74.3_{\pm0.1}$ | $77.4_{\pm0.6}$ | 66.3 |
| RSC | $97.5_{\pm0.6}$ | $63.1_{\pm1.2}$ | $73.0_{\pm1.3}$ | $76.2_{\pm0.5}$ | 77.5 | $60.7_{\pm1.4}$ | $51.4_{\pm0.3}$ | $74.8_{\pm1.1}$ | $75.1_{\pm1.3}$ | 65.5 |
| Mixup | $98.4_{\pm0.3}$ | $63.4_{\pm0.7}$ | $72.9_{\pm0.8}$ | $76.1_{\pm1.2}$ | 77.7 | $62.4_{\pm0.8}$ | $54.8_{\pm0.6}$ | $77.3_{\pm0.3}$ | $\textbf{79.2}_{\pm0.2}$ | 68.4 |
| DANN | $98.5_{\pm1.3}$ | $64.9_{\pm1.3}$ | $72.6_{\pm1.4}$ | $78.7_{\pm1.7}$ | 78.2 | $59.9_{\pm1.3}$ | $53.0_{\pm0.3}$ | $73.6_{\pm0.7}$ | $76.9_{\pm0.5}$ | 65.9 |
| CDANN | $97.6_{\pm0.6}$ | $65.2_{\pm0.8}$ | $73.4_{\pm1.4}$ | $76.9_{\pm0.5}$ | 78.3 | $61.5_{\pm1.4}$ | $50.4_{\pm2.4}$ | $74.4_{\pm0.9}$ | $76.6_{\pm0.8}$ | 65.8 |
| MTL | $97.6_{\pm0.6}$ | $60.6_{\pm1.3}$ | $71.0_{\pm1.2}$ | $77.2_{\pm0.7}$ | 76.6 | $61.5_{\pm0.7}$ | $52.4_{\pm0.6}$ | $74.9_{\pm0.4}$ | $76.8_{\pm0.4}$ | 66.4 |
| ARM | $97.2_{\pm0.5}$ | $62.7_{\pm1.}$ | $70.6_{\pm0.6}$ | $75.8_{\pm0.9}$ | 76.6 | $58.9_{\pm0.8}$ | $51.0_{\pm0.5}$ | $74.1_{\pm0.1}$ | $75.2_{\pm0.3}$ | 64.8 |
| IRM | $\textbf{98.6}_{\pm0.1}$ | $66.0_{\pm0.9}$ | $72.3_{\pm0.6}$ | $77.3_{\pm0.9}$ | 78.5 | $58.9_{\pm2.3}$ | $52.2_{\pm1.6}$ | $72.1_{\pm2.9}$ | $74.0_{\pm2.5}$ | 64.3 |
| SagNet | $97.3_{\pm0.4}$ | $61.6_{\pm0.8}$ | $73.4_{\pm1.9}$ | $77.6_{\pm0.4}$ | 77.5 | $\textbf{63.4}_{\pm0.2}$ | $54.8_{\pm0.4}$ | $75.8_{\pm0.4}$ | $78.3_{\pm0.3}$ | 68.1 |
| CRLII | $97.3_{\pm0.2}$ | $\textbf{67.2}_{\pm0.1}$ | $73.0_{\pm0.2}$ | $\textbf{78.8}_{\pm0.1}$ | **79.1** | $63.1_{\pm0.1}$ | $\textbf{56.9}_{\pm0.2}$ | $\textbf{78.8}_{\pm0.2}$ | $79.1_{\pm0.1}$ | **69.5** |

To evaluate our CRLII method more comprehensively, we applied it to more challenging datasets comprised of images sourced from the real world.

As shown in Table 2, our CRLII achieves superior OOD prediction performance on the PACS dataset, surpassing the next best by a margin of $2.3\%$. Additionally, it outperforms SOTA methods in the *Art painting*, *Cartoon*, and *Sketch* domains. We visualize the GradCAM of ERM, SageNet, and CRLII in Figure 4 and observe that our CRLII captures more invariant, discriminative features of the object. Our method also attains optimal average performance on both the VLCS and OfficeHome datasets, though it only surpasses SOTA methods in 2 out of the 4 domains for these datasets. The effectiveness of our CRLII method relies on how a given data distribution aligns with our assumptions. Empirical results underscore the capability of our proposed method to diminish spurious correlations and deliver robust, generalizable predictive performance across various applications.

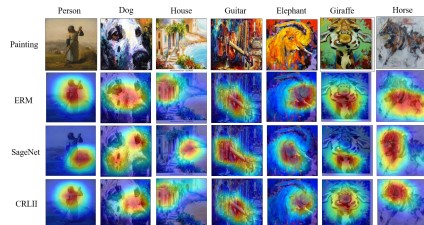

Figure 4: Visualization of Grad-CAMs on PACS dataset

## 6 CONCLUSION

In conclusion, our proposed framework for causal representation learning and interventional inference effectively disentangles robust and stable causal representations from spurious ones. This not only offers an interpretable explanation for causal representation in images but also enhances out-of-distribution prediction performance by estimating the interventional cause-effect between inputs and their corresponding labels. Empirical results further demonstrate that our approach surpasses SOTA domain generalization methods on benchmark distribution shift datasets.

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

## A   PROOF FOR THEOREM 3.1

*Proof.* To estimate $p\big(y|\boldsymbol{x}, do(\boldsymbol{z}_s)\big)$, we introduce variables $\boldsymbol{Z}_c$:

$$p\big(y|\boldsymbol{x}, do(\boldsymbol{z}_s)\big) = \int_{\boldsymbol{z}_c} p\big(y|\boldsymbol{z}_c, \boldsymbol{x}, do(\boldsymbol{z}_s)\big)p\big(\boldsymbol{z}_c|\boldsymbol{x}, do(\boldsymbol{z}_s)\big)\,\mathrm{d}\boldsymbol{z}_c \tag{7}$$

We can simplify the calculation of $p(y|\boldsymbol{z}_c, \boldsymbol{x}, do(\boldsymbol{z}_s))$ using Pearl's Do-Calculus Rules.

$$\begin{aligned}
p\big(y|\boldsymbol{z}_c, \boldsymbol{x}, do(\boldsymbol{z}_s)\big) =&p\big(y|\boldsymbol{z}_c, do(\boldsymbol{z}_s)\big) \quad (Y \perp\!\!\!\perp \boldsymbol{X}|\boldsymbol{Z}_c, \boldsymbol{Z}_s)_{\mathcal{G}_{\overline{\boldsymbol{Z}_s}}} \quad \text{According to Rule 1} \\
=&p\big(y|\boldsymbol{z}_c\big) \quad (Y \perp\!\!\!\perp \boldsymbol{Z}_s|\boldsymbol{Z}_c)_{\mathcal{G}_{\overline{\boldsymbol{Z}_s}}} \quad \text{According to Rule 3}
\end{aligned} \tag{8}$$

We employ the Backdoor Adjustment Theorem and Pearl's Do-Calculus Rule 2 to estimate $p\big(\boldsymbol{z}_c|\boldsymbol{x}, do(\boldsymbol{z}_s)\big)$,

$$\begin{aligned}
p\big(\boldsymbol{z}_c|\boldsymbol{x}, do(\boldsymbol{z}_s)\big) =&\frac{p\big(\boldsymbol{z}_c|do(\boldsymbol{z}_s)\big)p\big(\boldsymbol{x}|\boldsymbol{z}_s, do(\boldsymbol{z}_s)\big)}{p\big(\boldsymbol{x}|do(\boldsymbol{z}_s)\big)} \quad \text{Bayes's Theorem} \\
=&\frac{p\big(\boldsymbol{z}_s|do(\boldsymbol{z}_s)\big)p\big(\boldsymbol{x}|\boldsymbol{z}_c, \boldsymbol{z}_s\big)}{p\big(\boldsymbol{x}|do(\boldsymbol{z}_s)\big)} \quad (\boldsymbol{X} \perp\!\!\!\perp \boldsymbol{Z}_s|\boldsymbol{Z}_c)_{\mathcal{G}_{\underline{\boldsymbol{Z}_s}}} \quad \text{According to Rule 2}
\end{aligned} \tag{9}$$

Between $\boldsymbol{Z}_s, \boldsymbol{Z}_c$, there is a valid backdoor path from $\boldsymbol{Z}_c \leftarrow Y \leftarrow \boldsymbol{U}_{xy} \rightarrow \boldsymbol{Z}_s$, we can directly apply the Backdoor Adjustment Theorem with a valid adjusting set $\{Y\}$:

$$\begin{aligned}
p\big(\boldsymbol{z}_c|do(\boldsymbol{z}_s)\big) &= \sum_y p(\boldsymbol{z}_c|y, \boldsymbol{z}_s)p(y) \\
&= \sum_y p(\boldsymbol{z}_c|y)p(y) \quad (\boldsymbol{Z}_c \perp\!\!\!\perp \boldsymbol{Z}_s|Y)_{\mathcal{G}} \\
&= p(\boldsymbol{z}_c)
\end{aligned} \tag{10}$$

Between $\boldsymbol{Z}_s, \boldsymbol{X}$, there is a valid backdoor path from $\boldsymbol{X} \leftarrow \boldsymbol{Z}_c \leftarrow Y \leftarrow \boldsymbol{U}_{xy} \rightarrow \boldsymbol{Z}_s$. We are able to adjust on $\{Y\}$, $\{\boldsymbol{Z}_c\}$ or $\{Y, \boldsymbol{Z}_c\}$[3]. In our case, we choose to adjust on $\{\boldsymbol{Z}_c\}$:

$$p\big(\boldsymbol{x}|do(\boldsymbol{z}_s)\big) \overset{\text{Adjust on } \boldsymbol{Z}_c}{=} \int_{\boldsymbol{z}_c} p(\boldsymbol{x}|\boldsymbol{z}_c, \boldsymbol{z}_s)p(\boldsymbol{z}_c)\,\mathrm{d}\boldsymbol{z}_c \tag{11}$$

Substitute Eq. equation 10, equation 11 into Eq. equation 9, we obtain:

$$p\big(\boldsymbol{z}_c|\boldsymbol{x}, do(\boldsymbol{z}_s)\big) = \frac{p(\boldsymbol{x}|\boldsymbol{z}_c, \boldsymbol{z}_s)p(\boldsymbol{z}_c)}{\int_{\boldsymbol{z}_c} p(\boldsymbol{x}|\boldsymbol{z}_c, \boldsymbol{z}_s)p(\boldsymbol{z}_c)\,\mathrm{d}\boldsymbol{z}_c} \tag{12}$$

Substitute Eq. equation 12 and equation 8 into Eq. equation 7, we obtain Eq. equation 1 in **Theorem 3.1**:

$$\begin{aligned}
p\big(y|\boldsymbol{x}, do(\boldsymbol{z}_s)\big) &= \int_{\boldsymbol{z}_c} p\big(y|\boldsymbol{z}_c, \boldsymbol{x}, do(\boldsymbol{z}_s)\big)p\big(\boldsymbol{z}_c|\boldsymbol{x}, do(\boldsymbol{z}_s)\big)\,\mathrm{d}\boldsymbol{z}_c \\
&= \int_{\boldsymbol{z}_c} p(y|\boldsymbol{z}_c)\frac{p(\boldsymbol{x}|\boldsymbol{z}_c, \boldsymbol{z}_s)p(\boldsymbol{z}_c)}{\int_{\boldsymbol{z}_c} p(\boldsymbol{x}|\boldsymbol{z}_c, \boldsymbol{z}_s)p(\boldsymbol{z}_c)\,\mathrm{d}\boldsymbol{z}_c}\,\mathrm{d}\boldsymbol{z}_c
\end{aligned} \tag{13}$$

$\square$

---

[3]The equations for conditioning on those three different adjusting sets are the same.

## B DERIVATIONS

### B.1 THE DERIVATION IN EQ. (2)

$$
\begin{aligned}
\mathbb{E}_{p(\boldsymbol{z}_s|\boldsymbol{x})}[p(y|\boldsymbol{x}, do(\boldsymbol{z}_s))] =& \mathbb{E}_{p(\boldsymbol{z}_s|\boldsymbol{x})}\Big[\frac{\int_{\boldsymbol{z}_c} p(y|\boldsymbol{z}_c)p(\boldsymbol{x}|\boldsymbol{z}_c, \boldsymbol{z}_s)p(\boldsymbol{z}_c)\,\mathrm{d}\boldsymbol{z}_c}{\int_{\boldsymbol{z}_c} p(\boldsymbol{x}|\boldsymbol{z}_c, \boldsymbol{z}_s)p(\boldsymbol{z}_c)\,\mathrm{d}\boldsymbol{z}_c}\Big] \\
\approx& \frac{1}{N}\sum_{n=1}^{N}\Big[\sum_{l=1}^{L} p(y|\boldsymbol{z}_{c,l})\frac{p(\boldsymbol{x}|\boldsymbol{z}_{c,l}, \boldsymbol{z}_{s,n})}{\sum_{l'=1}^{L} p(\boldsymbol{x}|\boldsymbol{z}_{c,l'}, \boldsymbol{z}_{s,n})}\Big], \boldsymbol{z}_{c,l}\sim p(\boldsymbol{z}_c), \boldsymbol{z}_{s,n}\sim p(\boldsymbol{z}_s|\boldsymbol{x}) \\
=& \frac{1}{N}\sum_{n=1}^{N}\Big[\sum_{l=1}^{L} p(y|\boldsymbol{z}_{c,l})\boldsymbol{\omega}(\boldsymbol{z}_{c,l}, \boldsymbol{z}_{s,n})\Big], \text{where } \boldsymbol{\omega}(\boldsymbol{z}_{c,l}, \boldsymbol{z}_{s,n}) = \frac{p(\boldsymbol{x}|\boldsymbol{z}_{c,l}, \boldsymbol{z}_{s,n})}{\sum_{l'=1}^{L} p(\boldsymbol{x}|\boldsymbol{z}_{c,l'}, \boldsymbol{z}_{s,n})}
\end{aligned}
\tag{14}
$$

### B.2 THE DERIVATION OF THE MARGINAL LIKELIHOOD IN EQ. (4)

We character the joint likelihood over the variables in the proposed SCM. However, there are 6 variables of interest and only the observations for 2 of them are available. Therefore, we start from the marginal likelihood $p(\boldsymbol{x}, y)$.

$$
\begin{aligned}
&\log p(\boldsymbol{x}, y) \\
=& \log \int_{\boldsymbol{z}_c}\int_{\boldsymbol{z}_s}\sum_{u_x}\sum_{u_{xy}} p(u_x, u_{xy}, \boldsymbol{z}_s, \boldsymbol{z}_c, \boldsymbol{x}, y)\,\mathrm{d}\boldsymbol{z}_c\,\mathrm{d}\boldsymbol{z}_s \\
=& \log \int_{\boldsymbol{z}_c}\int_{\boldsymbol{z}_s}\sum_{u_x}\sum_{u_{xy}} p(u_x)p(u_{xy})p(\boldsymbol{z}_s|u_x, u_{xy})p(y|u_{xy})p(\boldsymbol{z}_c|y)p(\boldsymbol{x}|\boldsymbol{z}_s, \boldsymbol{z}_c)\,\mathrm{d}\boldsymbol{z}_c\,\mathrm{d}\boldsymbol{z}_s \quad \text{Bayesian Network Chain Rule} \\
=& \log \frac{1}{p(y)}\int_{\boldsymbol{z}_c}\int_{\boldsymbol{z}_s}\sum_{u_x}\sum_{u_{xy}} p(u_x)p(u_{xy})p(\boldsymbol{z}_s|u_x, u_{xy})p(y|u_{xy})p(y|\boldsymbol{z}_c)p(\boldsymbol{z}_c)p(\boldsymbol{x}|\boldsymbol{z}_s, \boldsymbol{z}_c)\,\mathrm{d}\boldsymbol{z}_c\,\mathrm{d}\boldsymbol{z}_s \quad \text{Bayes Theorem} \\
=& \log \frac{1}{p(y)} + \log \int_{\boldsymbol{z}_c}\int_{\boldsymbol{z}_s}\sum_{u_x}\sum_{u_{xy}} p(u_x)p(u_{xy})p(\boldsymbol{z}_s|u_x, u_{xy})p(y|u_{xy})p(y|\boldsymbol{z}_c)p(\boldsymbol{z}_c)p(\boldsymbol{x}|\boldsymbol{z}_s, \boldsymbol{z}_c)\,\mathrm{d}\boldsymbol{z}_c\,\mathrm{d}\boldsymbol{z}_s \\
\geq& \log \int_{\boldsymbol{z}_c}\int_{\boldsymbol{z}_s}\Big[\sum_{u_x}\sum_{u_{xy}} p(u_x)p(u_{xy})p(\boldsymbol{z}_s|u_x, u_{xy})p(y|u_{xy})\Big]p(y|\boldsymbol{z}_c)p(\boldsymbol{z}_c)p(\boldsymbol{x}|\boldsymbol{z}_s, \boldsymbol{z}_c)\,\mathrm{d}\boldsymbol{z}_c\,\mathrm{d}\boldsymbol{z}_s \\
=& \log \int_{\boldsymbol{z}_c}\int_{\boldsymbol{z}_s}\Big[\sum_{u_x}\sum_{u_{xy}} p(u_x|u_{xy})p(u_{xy})p(\boldsymbol{z}_s|u_x, u_{xy})p(y|u_{xy})\Big]p(y|\boldsymbol{z}_c)p(\boldsymbol{z}_c)p(\boldsymbol{x}|\boldsymbol{z}_s, \boldsymbol{z}_c)\,\mathrm{d}\boldsymbol{z}_c\,\mathrm{d}\boldsymbol{z}_s \quad U_x \perp\!\!\!\perp U_{xy} \\
=& \log \int_{\boldsymbol{z}_c}\int_{\boldsymbol{z}_s}\frac{\big[\sum_{u_{xy}} p(u_{xy})p(\boldsymbol{z}_s|u_{xy})p(y|u_{xy})\big]p(y|\boldsymbol{z}_c)p(\boldsymbol{z}_c)p(\boldsymbol{x}|\boldsymbol{z}_s, \boldsymbol{z}_c)}{q(\boldsymbol{z}_s, \boldsymbol{z}_c|\boldsymbol{x})}q(\boldsymbol{z}_s, \boldsymbol{z}_c|\boldsymbol{x})\,\mathrm{d}\boldsymbol{z}_c\,\mathrm{d}\boldsymbol{z}_s \\
=& \log \mathbb{E}_{q(\boldsymbol{z}_s, \boldsymbol{z}_c|\boldsymbol{x})}\Big[\frac{\big[\sum_{u_{xy}} p(u_{xy})p(\boldsymbol{z}_s|u_{xy})p(y|u_{xy})\big]p(y|\boldsymbol{z}_c)p(\boldsymbol{z}_c)p(\boldsymbol{x}|\boldsymbol{z}_s, \boldsymbol{z}_c)}{q(\boldsymbol{z}_s, \boldsymbol{z}_c|\boldsymbol{x})}\Big] \\
=& \log \mathbb{E}_{q(\boldsymbol{z}_s, \boldsymbol{z}_c|\boldsymbol{x})}\Big[\frac{\big[\sum_{u_{xy}} p(u_{xy})p(\boldsymbol{z}_s|u_{xy})p(y|u_{xy})\big]p(y|\boldsymbol{z}_c)p(\boldsymbol{z}_c)p(\boldsymbol{x}|\boldsymbol{z}_s, \boldsymbol{z}_c)}{q(\boldsymbol{z}_s|\boldsymbol{x})q(\boldsymbol{z}_c|\boldsymbol{x})}\Big] \quad \text{Assume } q(\boldsymbol{z}_s, \boldsymbol{z}_c|\boldsymbol{x}) = q(\boldsymbol{z}_s|\boldsymbol{x})q(\boldsymbol{z}_c|\boldsymbol{x}) \\
\geq& \mathbb{E}_{q(\boldsymbol{z}_s, \boldsymbol{z}_c|\boldsymbol{x})}\log\Big[\frac{\big[\sum_{u_{xy}} p(u_{xy})p(\boldsymbol{z}_s|u_{xy})p(y|u_{xy})\big]p(\boldsymbol{z}_c)}{q(\boldsymbol{z}_s|\boldsymbol{x})q(\boldsymbol{z}_c|\boldsymbol{x})}p(y|\boldsymbol{z}_c)p(\boldsymbol{x}|\boldsymbol{z}_s, \boldsymbol{z}_c)\Big] \quad \text{Jensen's inequality} \\
=& \mathbb{E}_{q(\boldsymbol{z}_s, \boldsymbol{z}_c|\boldsymbol{x})}\Big[\log\frac{\big[\sum_{u_{xy}} p(u_{xy})p(\boldsymbol{z}_s|u_{xy})p(y|u_{xy})\big]}{q(\boldsymbol{z}_s|\boldsymbol{x})} + \log\frac{p(\boldsymbol{z}_c)}{q(\boldsymbol{z}_c|\boldsymbol{x})} + \log p(y|\boldsymbol{z}_c) + \log p(\boldsymbol{x}|\boldsymbol{z}_s, \boldsymbol{z}_c)\Big] \\
=& \mathbb{E}_{q(\boldsymbol{z}_s|\boldsymbol{x})}\log\frac{\sum_{u_{xy}} p(u_{xy})p(\boldsymbol{z}_s|u_{xy})p(y|u_{xy})}{q(\boldsymbol{z}_s|\boldsymbol{x})} + \mathbb{E}_{q(\boldsymbol{z}_c|\boldsymbol{x})}\log\frac{p(\boldsymbol{z}_c)}{q(\boldsymbol{z}_c|x)} + \mathbb{E}_{q(\boldsymbol{z}_c|\boldsymbol{x})}\log p(y|\boldsymbol{z}_c) + \mathbb{E}_{q(\boldsymbol{z}_c, \boldsymbol{z}_s|\boldsymbol{x})}\log p(\boldsymbol{x}|\boldsymbol{z}_c, \boldsymbol{z}_s) \\
=& \mathbb{E}_{q(\boldsymbol{z}_s|\boldsymbol{x})}\log\frac{\sum_{u_{xy}} p(u_{xy})p(\boldsymbol{z}_s|u_{xy})p(y|u_{xy})}{q(\boldsymbol{z}_s|\boldsymbol{x})} - KL\big(q(\boldsymbol{z}_c|\boldsymbol{x})||p(\boldsymbol{z}_c)\big) + \mathbb{E}_{q(\boldsymbol{z}_c|\boldsymbol{x})}\log p(y|\boldsymbol{z}_c) + \mathbb{E}_{q(\boldsymbol{z}_c, \boldsymbol{z}_s|\boldsymbol{x})}\log p(\boldsymbol{x}|\boldsymbol{z}_c, \boldsymbol{z}_s)
\end{aligned}
\tag{15}
$$

We further simplify the first term as follows,

$$
\mathbb{E}_{q(\boldsymbol{z}_s|\boldsymbol{x})} \log \frac{\left[\sum_{u_{xy}} p(u_{xy}) p(\boldsymbol{z}_s|u_{xy}) p(y|u_{xy})\right]}{q(\boldsymbol{z}_s|\boldsymbol{x})}
$$

$$
= \mathbb{E}_{q(\boldsymbol{z}_s|\boldsymbol{x})} \log \left[\sum_{u_{xy}} \frac{p(\boldsymbol{z}_s|u_{xy}) p(y|u_{xy})}{q(\boldsymbol{z}_s|\boldsymbol{x})} p(u_{xy})\right]
$$

$$
= \mathbb{E}_{q(\boldsymbol{z}_s|\boldsymbol{x})} \log \mathbb{E}_{p(u_{xy})} \frac{p(\boldsymbol{z}_s|u_{xy}) p(y|u_{xy})}{q(\boldsymbol{z}_s|\boldsymbol{x})}
$$

$$
\geq \mathbb{E}_{q(\boldsymbol{z}_s|\boldsymbol{x})} \mathbb{E}_{p(u_{xy})} \log \left[\frac{p(\boldsymbol{z}_s|u_{xy}) p(y|u_{xy})}{q(\boldsymbol{z}_s|\boldsymbol{x})}\right] \quad \text{Jensen's inequality}
$$

$$
= \mathbb{E}_{q(\boldsymbol{z}_s|\boldsymbol{x})} \mathbb{E}_{p(u_{xy})} \left[\log \frac{p(\boldsymbol{z}_s|u_{xy})}{q(\boldsymbol{z}_s|\boldsymbol{x})} + \log p(y|u_{xy})\right]
$$

$$
= \mathbb{E}_{p(u_{xy})} \mathbb{E}_{q(\boldsymbol{z}_s|\boldsymbol{x})} \left[\log \frac{p(\boldsymbol{z}_s|u_{xy})}{q(\boldsymbol{z}_s|\boldsymbol{x})}\right] + \mathbb{E}_{p(u_{xy})} \log p(y|u_{xy})
$$

$$
= \mathbb{E}_{p(u_{xy})} \mathbb{E}_{q(\boldsymbol{z}_s|\boldsymbol{x})} \left[\log \frac{\sum_{u_x} p(\boldsymbol{z}_s|u_x, u_{xy}) p(u_x|u_{xy})}{q(\boldsymbol{z}_s|\boldsymbol{x})}\right] + \mathbb{E}_{p(u_{xy})} \log p(y|u_{xy}) \ \text{Re-introduce } U_x
$$

$$
= \mathbb{E}_{p(u_{xy})} \mathbb{E}_{q(\boldsymbol{z}_s|\boldsymbol{x})} \left[\log \sum_{u_x} \frac{p(\boldsymbol{z}_s|u_x, u_{xy})}{q(\boldsymbol{z}_s|\boldsymbol{x})} p(u_x)\right] + \mathbb{E}_{p(u_{xy})} \log p(y|u_{xy}) \quad U_x \perp\!\!\!\perp U_{xy}
$$

$$
= \mathbb{E}_{p(u_{xy})} \mathbb{E}_{q(\boldsymbol{z}_s|\boldsymbol{x})} \left[\log \mathbb{E}_{p(u_x)} \frac{p(\boldsymbol{z}_s|u_x, u_{xy})}{q(\boldsymbol{z}_s|\boldsymbol{x})}\right] + \mathbb{E}_{p(u_{xy})} \log p(y|u_{xy})
$$

$$
\geq \mathbb{E}_{p(u_{xy})} \mathbb{E}_{p(u_x)} \mathbb{E}_{q(\boldsymbol{z}_s|\boldsymbol{x})} \left[\log \frac{p(\boldsymbol{z}_s|u_x, u_{xy})}{q(\boldsymbol{z}_s|\boldsymbol{x})}\right] + \mathbb{E}_{p(u_{xy})} \log p(y|u_{xy}) \quad \text{Jensen's inequality}
$$

$$
= -\mathbb{E}_{p(u_x)} \mathbb{E}_{p(u_{xy})} KL\big(q(\boldsymbol{z}_s|\boldsymbol{x})||p(\boldsymbol{z}_s|u_x, u_{xy})\big) + \mathbb{E}_{p(u_{xy})} \log p(y|u_{xy})
$$

$$
= -\mathbb{E}_{p(u_x, u_{xy})} KL\big(q(\boldsymbol{z}_s|\boldsymbol{x})||p(\boldsymbol{z}_s|u_x, u_{xy})\big) + \mathbb{E}_{p(u_{xy})} \log p(y|u_{xy})
$$

$$
\tag{16}
$$

We parameterize the encoder distributions using parameter $\Theta = \{\Theta_s, \Theta_c\}$, denoted as $q_{\Theta_s}(\boldsymbol{z}_s|\boldsymbol{x})$ and $q_{\Theta_c}(\boldsymbol{z}_c|\boldsymbol{x})$, the decoder distribution with parameter $\Phi$ as $p_\Phi(\boldsymbol{x}|\boldsymbol{z}_c, \boldsymbol{z}_s)$, and the classifier distribution with parameter $\Psi$ as $p_\Psi(y|\boldsymbol{z}_c)$. During training, we optimize these defined parameters to construct the corresponding distributions. Additionally, we make assumptions or estimations about the prior distributions, specifically $p(\boldsymbol{z}_c)$ and $p(\boldsymbol{z}_s|u_x, u_{xy})$, to help regularize the learning of representations. Notably, we do not parameterize over $p(y|u_{xy})$, as it is not necessary for either obtaining representations or computing interventional distributions. As a result, we omit the term $\mathbb{E}_{p(u_{xy})} \log p(y|u_{xy})$, as it is independent of the parameters for optimization. By combining Eq.(15) with Eq.(16), we propose the following training objective for the causal representation learning procedure:

$$
\mathcal{L}_{obj}(\boldsymbol{x}, y, \Theta, \Phi, \Psi) = -\mathbb{E}_{p(u_x, u_{xy})} KL\big(q_{\Theta_s}(\boldsymbol{z}_s|\boldsymbol{x})||p(\boldsymbol{z}_s|u_x, u_{xy})\big) - KL\big(q_{\Theta_c}(\boldsymbol{z}_c|\boldsymbol{x})||p(\boldsymbol{z}_c)\big)
$$
$$
+ \mathbb{E}_{q_{\Theta_c}(\boldsymbol{z}_c|\boldsymbol{x})} \log p_\Psi(y|\boldsymbol{z}_c) + \mathbb{E}_{q_\Theta(\boldsymbol{z}_c, \boldsymbol{z}_s|\boldsymbol{x})} \log p_\Phi(\boldsymbol{x}|\boldsymbol{z}_c, \boldsymbol{z}_s)
$$

$$
\tag{17}
$$

## C Algorithm for Inference

---

**Algorithm 2** Interventional Inference

---

1: **Input:** An input from test set $\boldsymbol{x}^t$; trained encoders $q_{\hat{\Theta}_c}(\boldsymbol{Z}_c|\boldsymbol{X})$ and $q_{\hat{\Theta}_s}(\boldsymbol{Z}_s|\boldsymbol{X})$; trained classifier $p_{\hat{\Psi}}(Y|\boldsymbol{Z}_c)$; Training data $\mathcal{D}$.
2: **for** $n = 1, 2, \cdots, N$ **do**
3:     Sample $\boldsymbol{z}_{s,n}^t \sim q_{\hat{\Theta}_s}(\boldsymbol{z}_s|\boldsymbol{x}^t)$.
4:     **for** $\boldsymbol{x}^i \sim \mathcal{D}$ **do**
5:         Sample $\boldsymbol{z}_{c,i} \sim q_{\hat{\Theta}_c}(\boldsymbol{z}_c|\boldsymbol{x}^i)$. Compute $\boldsymbol{w}(\boldsymbol{z}_{c,i}, \boldsymbol{z}_{s,n}^t)$.
6:         Select $L$ samples of $\boldsymbol{z}_{c,l}$ with the highest $\boldsymbol{w}(\boldsymbol{z}_{c,l}, \boldsymbol{z}_{s,n}^t)$ and their corresponding $\boldsymbol{x}^l$.
7:     **end for**
8:     Given $\{\boldsymbol{z}_{c,l}, \boldsymbol{x}^l\}_{l=1}^L$, Compute $\{p(y|\boldsymbol{z}_{c,l}), \boldsymbol{w}(\boldsymbol{z}_{c,l}, \boldsymbol{z}_{s,n}^t)\}_{l=1}^L$ for $\boldsymbol{Z}_s = \boldsymbol{z}_{s,n}^t$.
9: **end for**
10: Compute $\mathbb{E}_{p(\boldsymbol{z}_s|\boldsymbol{x}^t)}[p(y|\boldsymbol{x}^t, do(\boldsymbol{z}_s))]$ via Eq. (6)
11: $\hat{y}^t \leftarrow \arg\max_y \mathbb{E}_{p(\boldsymbol{z}_s|\boldsymbol{x}^t)}[p(y|\boldsymbol{x}^t, do(\boldsymbol{z}_s))]$
12: **Output** Prediction of label: $\hat{y}^t$

---

## D Assumptions

**Theorem and practical gap:** We would like to emphasize that the causal mechanisms in the proposed SCM in Figure 1 are all assumptions that are widely adopted in the area of causal representation learning, including the three following points: 1) The latent high-level factors $\boldsymbol{Z}$ can be separated into causal factors $\boldsymbol{Z}_c$ and spurious factors $\boldsymbol{Z}_s$. 2) The input $\boldsymbol{X}$ is generated by the the high-level factors $\boldsymbol{Z}$. 3) Causal factor $\boldsymbol{Z}_c$ is either direct cause or effect of target $Y$. 4) The latent confounding effects between $\boldsymbol{Z}_s$ and $Y$ remain invariant. 5) The domain specific factor $U_x$ varies across domains. We cannot prove these assumptions always hold in real-world data and we admit the effectiveness and soundness of our derived theorem and algorithm is built upon these assumptions. This framework may not generalize to new domains with different and unknown confounding effects.

**Factorized variational distribution $q$:** While we acknowledge that the true distribution $p(\boldsymbol{Z}_c, \boldsymbol{Z}_s|\boldsymbol{X})$ cannot be factored under our SCM assumption, it often becomes intractable due to the conditions that only $X$ and $Y$ are observed in the SCM. The distribution $p(\boldsymbol{Z}_c, \boldsymbol{Z}_s|\boldsymbol{X})$ is likely to be complex and non-Gaussian. As an alternative solution, we approximate $p(\boldsymbol{Z}_c, \boldsymbol{Z}_s|\boldsymbol{X})$ by a variational distribution $q(\boldsymbol{Z}_c, \boldsymbol{Z}_s|\boldsymbol{X})$ with Gaussian assumptions. Although using non-factorized Gaussians might slightly enhance results, it requires more time to estimate the large covariance matrix. There is always a balance between accuracy and efficiency. We believe our assumption, commonly used in many VAEs, is reasonable and effective. Moreover, our empirical results indicate that the factorized Gaussian approximation effectively leads to better OOD generalization. We appreciate your suggestion and will include an analysis of using non-factorized Gaussians in our revised paper.

## E Empirical Ablation Study

In this section, we perform three ablation studies: 1) We show the sensitivity of our proposed CRLII method with respect to the value of $|U|$ that we specify during SCM parameterization and learning. 2) We demonstrate the necessity of our intervention inference approach due to the imperfect disentanglement between $\boldsymbol{z}_s$ and $\boldsymbol{z}_c$. 3) We provide a detailed approach to choose the number of $\boldsymbol{z}_{c,l}$ that we need to obtain to perform interventional inference.

### E.1 The number of domains

In this section, we explore how varying the number of domains $|U|$ affects out-of-distribution (OOD) prediction performance. We incrementally increase the values of $|U|$ from 1 to 5 and present the corresponding prediction performance in Figure 5.

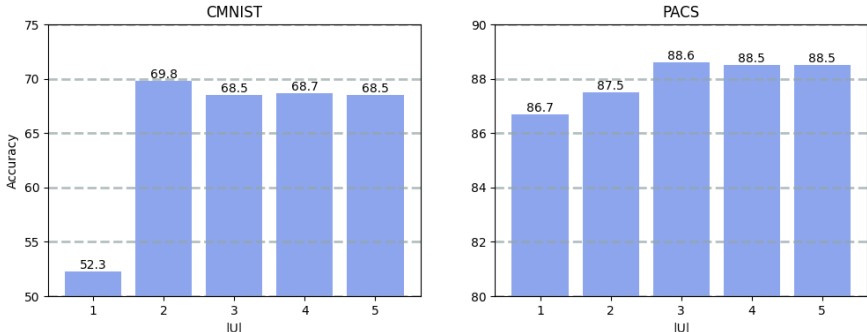

Figure 5: Ablation study on the influence from the number of domains $|U|$. The results on PACS dataset are averaged over four domains.

The findings from Figure 5 reveal that our method exhibits its poorest performance when the number of domains, denoted as $|U|$, is set to 1. In such a scenario, the assumption is made that $z_s$ follows a Gaussian distribution, and its prior distribution mirrors that of $z_c$. This results in a compromise in the asymmetry regularization between the two types of representations, leading to suboptimal disentanglement. The restoration of this asymmetry occurs when we set $|U| \geq 2$. However, performance results appear comparable for cases where $|U|$ exceeds 2. The optimal selection of $|U|$ hinges on the dissimilarities between the true data distributions across each domain. In situations involving observational datasets lacking domain-specific information, setting $|U| = 2$ can still yield a reasonably well-disentangled set of representations.

## E.2 Influence of interventional inference

In Section 4.1, we provide theoretical justification for our choice of interventional inference, driven by the partial disentanglement observed between $z_c$ and $z_s$ during the SCM learning process. In Table 4, our empirical results demonstrate that the $z_c$ representation we obtain still retains information from $z_s$. Consequently, interventional inference proves effective in further enhancing OOD performance when compared to direct prediction using $p(y|z_c^t)$, where $z_c^t = \arg\max_{z_c} q(z_c|x^t)$.

Table 4: Comparison between prediction and interventional inference.

| Datasets | Accuracy (%) | |
| --- | --- | --- |
| | Prediction with $z_c$ | Interventional Inference |
| CMNIST | 52.6 | **69.8** |
| PACS | 86.7 | **88.6** |
| VLCS | 76.3 | **79.1** |
| OfficeHome | 67.7 | **69.5** |

## E.3 The selection of $L$

For the purpose of inference, we generated a set of $z_c$ samples from the training inputs and computed their weighted sum for $p(y|z_c)$. However, this process proved to be time-consuming and inefficient, especially when dealing with a large number of training inputs. We observed significant variation in the magnitudes of weights assigned to different samples of $z_c$. Let's denote the obtained samples as $z_{c,1}, z_{c,2}, \cdots, z_{c,l}$, with $\omega(z_{c,1}, z_{s,n}) \geq \omega(z_{c,2}, z_{s,n}) \geq \cdots \geq \omega(z_{c,L}, z_{s,n})$. Notably, when $L > 5$, the ratio $\frac{\omega(z_{c,1}, z_{s,n})}{\omega(z_{c,L}, z_{s,n})}$ exceeds 10, and when $L > 10$, it surpasses 100. In cases where $\omega(z_{c,1}, z_{s,n}) > 100\omega(z_{c,L}, z_{s,n})$, the contribution of $\omega(z_{c,L}, z_{s,n})$ to the weighted sum becomes negligible. Consequently, it becomes unnecessary to consider values of $L$ greater than 10.

