# OpenReview forum: "Causal Representation Learning and Inference for Generalizable Cross-Domain Predictions"
_ICLR.cc/2024/Conference — Submitted to ICLR 2024_

### Official Review · Reviewer_Rf1b · 2023-10-30

**Soundness:** 2 fair
**Presentation:** 3 good
**Contribution:** 2 fair
**Rating:** 5
**Confidence:** 3

**Summary:**

The authors propose a domain generalization algorithm motivated by causality. They specify two latent variables $u_x$ and $u_{xy}$ whose marginals can shift between training and testing. Their approach aims to become invariant to the aforementioned latent variables by intervening on their common child $z_s$, which closes the backdoor path between the input and target.

**Strengths:**

The authors tackle the important problem of DG, and their approach shows promising empirical results. The paper is well-written and clearly motivated. Also, their approach is interesting in that unlike many existing DG algorithms, theirs doesn't use the environment labels.

**Weaknesses:**

I found three technical issues with the paper. One is a major issue, and two are minor.

1. (Major) The algorithm does not perform its stated purpose of being invariant to shifts in $u_x$ and $u_{xy}$. The predictive distribution in Eq. (2) is not invariant to $u_x$ and $u_{xy}$, since it involves an expectation over $p(z_s \mid x)$, which can shift across training and testing.

2. (Minor) The posterior is assumed to factorize $q(z_c, z_s \mid x) = q(z_c \mid x) q(z_s \mid x)$, which is at odds with the assumed causal graph.

3. (Minor) The authors cite Kivva 2022 to claim that their standard normal prior $p(z_c)$ is a one-component Gaussian mixture, and therefore $z_c$ is identifiable (along w/ the piecewise affine decoder assumption). Calling a standard normal distribution a Gaussian mixture is technically true, but this identifiability argument is a bit tenuous.

**Questions:**

Please address my three points above in the "weaknesses" section.

---

> ### Author Response · Authors · 2023-11-20
> **Responses to Reviewer Rf1b (part 1)**
>
> **Invariance of $p(Y|X, do(Z_s))$:** To address the reviewer’s concerns, we will first emphasize and rectify the assumptions we make regarding the latent variables $U_x, U_{xy}$. Then we will explain that under these assumptions, $p(Y|X, do(Z_s))$ is invariant and transportable across domains. Finally, we will explain how to obtain $Z_s$ values for calculating $p(Y|X, do(Z_s))$ and why it is a reasonable and good choice we have to approximate an invariant transformation between $Z_s$ and $X$.
>
> - **Assumptions:** Upon careful review, we acknowledge that our assertion regarding the variation of $p(U_{xy})$ was overstated. It is imperative to make the confounding effects between the source domain and target domain consistent to make the proposed interventional distribution invariant and transportable. Hence we correct our initial assumption that the confounding effects, encompassing $p(U_{xy})$ and the causal mechanisms between $U_{xy}$, $Z_s$, and $Y$, remain consistent across training and test domains. Moreover, we assume the distribution of $p(U_x)$ varies across domains. It results in the variation of $p(Z_s)$, and further the variation of $p(X)$. However, we emphasize that the generative mechanism $p(X|Z_s, Z_c)$ remain invariant across domains ($X$ is independent of $U_x, U_{xy}$ given $Z_c, Z_c$). Otherwise, it is impossible to infer $Z$ from any unseen domain.
>
> - **Invariance of $p(Y|X, do(Z_s))$:** Our proposed interventional distribution, by setting specific values for $Z_s$ to mitigate the influence of the domain-specific variable $U_x$ on $Z_s$, effectively addresses the invariant confounding effects and prevent the $U_x$ from influence $Y$. Hence, it is invariant and transportable across domains.
> - **The choice of $Z_s$:** Under our corrected assumption, the latent confounding effect remains invariant across domains. To infer the label for a test input $x^t$ using our interventional distribution, it is imperative to provide the true value of $Z_s$ for $x^t$ to accurately account for the confounding effects between $x^t$ and $y^t$. The distribution $p(Z_s|X)$ utilized in Eq. (3) can be construed as a proxy distribution that enables us to derive the true values of $Z_s$ from an input $X$. The challenge lies in determining which learned distribution can be employed to approximate this proxy distribution. According to the SCM, we observe that $X$ is independent of $U$ given $Z=[Z_c, Z_s]$, rendering $p(X|Z)$ invariant and transportable. Therefore, a reasonable and good choice is to obtain the identifiable $z_s^t$ from the learned variational distribution $q(Z_s|X=x^t)$ with a high $p(X=x^t|Z_s=z^t_s)$. However, as $q(Z_s|X=x^t)$ represents a variational estimation of the desired proxy distribution, we average the interventional distribution over multiple samples of $z_s$ from $q(Z_s|X=x^t)$. In practice, randomly obtaining a $z_s$ value is likely to yield a low $p(X|Z)$ and contribute minimally to the calculation of the interventional distribution.
>
> Nevertheless, it's crucial to underscore that with the corrected assumption, our CIIRL remains innovative and has proven its efficacy across various benchmark distribution shift datasets: 1) With the revised assumptions, our proposed interventional distribution maintains invariance and transportability, facilitating cross-domain inference. 2) Our training procedure does not explicitly rely on the variation of $p(U_{xy})$. The distinct classes of $U$ can be associated with the diverse domain variable $U_x$. Our training procedure can still effectively distinguish the two types of representations. 3) Empirical results affirm the existence of invariant latent confounders, as evidenced by the overall superior OOD prediction performance of CIIRL compared to predictions solely based on $Z_c$.

---

> ### Author Response · Authors · 2023-11-20
> **Responses to Reviewer Rf1b (part2)**
>
> **Factorized variational distribution $q$:** While we acknowledge that the true distribution $p(Z_c, Z_s|X)$ cannot be factored under our SCM assumption, it often becomes intractable due to the conditions that only $X$ and $Y$ are observed in the SCM.  The distribution $p(Z_c, Z_s|X)$ is likely to be complex and non-Gaussian. As an alternative solution, we approximate $p(Z_c, Z_s|X)$ by a variational distribution $q(Z_c, Z_s|X)$ with Gaussian assumptions. Although using non-factorized Gaussians might slightly enhance results, it requires more time to estimate the large covariance matrix. There is always a balance between accuracy and efficiency. We believe our assumption, commonly used in many VAEs, is reasonable and effective. Moreover, our empirical results indicate that the factorized Gaussian approximation effectively leads to better OOD generalization. We appreciate your suggestion and will include an analysis of using non-factorized Gaussians in our revised paper.
>
>
> **Identifiability:** We emphasize that our proof of identifiability, leveraging the results of [1], is mathematically correct. However, it is important to note that this proof is not our main contribution. Our focus is on designing conditional distributions in our SCM to ensure that the learned latent representations are identifiable. While Kivva et al. have proven identifiability in a variety of cases, our work necessitates only a specific instance to effectively learn our SCM. We model $p(Z_c)$ as a standard Gaussian instead of a mixture of Gaussian, chosen for its simplicity and because we lack additional prior information about $Z_c$. This assumption aligns with the prior distributions commonly used in many VAEs.
>
> > [1] Kivva, Bohdan, et al. "Identifiability of deep generative models without auxiliary information." Advances in Neural Information Processing Systems 35 (2022): 15687-15701.

---

### Official Review · Reviewer_kccw · 2023-11-01

**Soundness:** 2 fair
**Presentation:** 3 good
**Contribution:** 3 good
**Rating:** 5
**Confidence:** 3

**Summary:**

The work proposes a causal representation learning procedure for domain generalization given data from a single domain.  An invariance relation is derived based on interventions on the spurious representation. The proposed procedure aims to identify the latent causal and spurious representations and then make predictions according to the invariance relation.

**Strengths:**

1. The representation learning procedure is novel and interesting, especially the interventions on $Z_{s}$.

2. The method outperforms the baselines by a large margin on the CMNIST dataset.

**Weaknesses:**

1. The latent confounder $U_{xy}$ is assumed to be discrete, which is restrictive. The dependency between $Y$ and $Z_{s}$ can be more complicated in general.

2. The identifiability of the representation is a crucial result. From the discussion in Section 4.1, the identifiability results are not trivial. I think they should be written in a formal statement and proved rigorously.

3. A claim is that $p(Y|X,do(Z_{s}))$ is invariant across different distributions due to the removed arrows $U_x \to Z_{s}$ and $U_{xy} \to Y$. However, there is still an arrow $U_{xy} \to Y$, meaning that the marginal distribution of $Y$ can change across different distributions. As a result,  $p(Y|X,do(Z_{s}))$ is not invariant in general.

**Questions:**

1. Whether the assumption of a discrete  $U_{xy}$ can be relaxed? What are the consequences of a large $J=|U|$?

2. Does the confounder make the invariance fail as mentioned above?

I may raise my score depending on the response. If the invariance indeed fails, I would recommend rejection.

---

> ### Author Response · Authors · 2023-11-20
> **Responses to Reviewer kccw**
>
> **Assumptions of $U_{x}, U_{xy}$:** The assumptions we make about $U_x$ and $U_{xy}$ posit them as random variables representing domain-specific and confounding information, respectively. We adhere to the standard practice of simplifying $U_x$ into a domain index. Importantly, we do not constrain $U_x$ and $U_{xy}$ to be exclusively discrete or continuous. During the training process, a neural network is employed to take in $U_x$ and $U_{xy}$ and generate parameters, including the mean and variances, for the prior distributions of $Z_s$. The neural network accommodates inputs of any type. However, the training procedure disentangles $Z_s$ and $Z_c$ by employing asymmetric prior distributions. As the dimensionality $J=|U|$ increases, optimization becomes more challenging due to potential inaccuracies in estimating domain indices, a growing number of parameters in the prior distribution, and limited improvements (if any) in disentanglement.
>
> **Invariance of $p(Y|X, do(Z_s))$:** Upon careful review, we acknowledge that our assertion regarding the variation of $p(U_{xy})$ was overstated. Our proposed interventional distribution, by setting specific values for $Z_s$ to mitigate the influence of the domain-specific variable $U_x$ on $Z_s$, effectively accounts for the invariant confounding effects and prevents the $U_x$ from influencing $Y$. However, it is important to note that this framework may not generalize to new domains with different and unknown confounding effects.
>
> In light of this, we correct our initial assumption that the confounding effects, encompassing $p(U_{xy})$ and the causal mechanisms between $U_{xy}$, $Z_s$, and $Y$, remain consistent across training and test domains. This invariant confounding assumption aligns with standard practices widely adopted in works utilizing interventional distributions [1]. We will incorporate such a revision into the paper.
>
> Nevertheless, it's crucial to underscore that with the corrected assumption, our CIIRL remains innovative and has proven its efficacy across various benchmark distribution shift datasets:
> - 1) With the revised assumptions, our proposed interventional distribution maintains invariance and transportability, facilitating cross-domain inference.
> - 2) Our training procedure does not explicitly rely on the variation of $p(U_{xy})$. The distinct classes of $U$ can be associated with the diverse domain variable $U_x$. Our training procedure can still effectively distinguish the two types of representations.
> - 3) Empirical results affirm the existence of invariant latent confounders, as evidenced by the overall superior OOD prediction performance of CIIRL compared to predictions solely based on $Z_c$.
>
> > [1] Mao, Chengzhi, et al. "Causal transportability for visual recognition." Proceedings of the IEEE/CVF Conference on Computer Vision and Pattern Recognition. 2022.
>
> **Identifiability:** We emphasize that our proof of identifiability, leveraging the results of [2], is mathematically correct. However, it is important to note that this proof is not our main contribution. However, we appreciate the suggestion of the reviewer and will consider dedicating a formal statement with rigorous proof for identifiability.
>
> > [2] Kivva, Bohdan, et al. "Identifiability of deep generative models without auxiliary information." Advances in Neural Information Processing Systems 35 (2022): 15687-15701.

---

### Official Review · Reviewer_ni4b · 2023-11-01

**Soundness:** 1 poor
**Presentation:** 3 good
**Contribution:** 3 good
**Rating:** 3
**Confidence:** 4

**Summary:**

This paper aims to solve the problem of out-of-distribution classification using a causal approach. In the problem setting, the features $X$ are caused by causal latent variables $Z_C$ and spurious latent variables $Z_S$ and are correlated with labels $Y$ through both sets of latent variables. A typical classifier predicts $P(Y \mid X)$, using the correlation through both $Z_S$ and $Z_C$. However, under distribution shift, the distribution of unobserved variables affecting $Z_S$ are changed, so using the spurious latent variables for classification can result in incorrect predictions out-of-distribution. Instead, the paper proposes using $P(Y \mid X, do(Z_S))$ for classification, which severs the correlation between $Y$ and $X$ through $Z_S$ via a causal intervention, thus providing a quantity that is invariant across domains. Estimating this quantity requires learning encoders which map $X$ to $Z_C$ and $Z_S$, a decoder which maps $Z_S$ and $Z_C$ back to $X$, and a classifier $P(Y \mid Z_C)$. This is done by optimizing over a variational bound on the log-likelihood of the data. After training, predictions are obtained by computing a linear combination of predictions from $P(Y \mid Z_C)$ weighted by a value indicating the compatibility of $Z_C$ with $X$ (using Monte Carlo sampling to estimate expectations). Experiments demonstrate the effectiveness of the approach.

**Strengths:**

This paper offers a novel take on leveraging causality to solve out-of-distribution classification. To my knowledge, there are no works which consider modeling the problem as done in Fig. 1, where $P(y \mid x, do(z_S))$ is used as the classifier. The problem setup has interesting implications in terms of the ways that features $X$ and label $Y$ are related. The experimental results also show promise that the approach is effective in practice.

**Weaknesses:**

I am concerned about the soundness of some of the claims:

1. The path from $U_{xy}$ to $Y$ is not influenced by any intervention on $Z_s$. Hence, if $p^s(U_{xy}) \neq p^t(U_{xy})$, it should also be the case that $p^s(y \mid x, do(z_s)) \neq p^t(y \mid x, do(z_s))$. This seems to contradict what is stated at the end of Sec. 3.1.

2. It is not clear how calculating the expectation of $p(y \mid x, do(z_s))$ over $p(z_s \mid x)$ (as done so in Eq. 2) is considered marginalizing out $z_s$. It is also not clear why this is preferable to just choosing some arbitrary $z_s$ to intervene.

3. How are $p(u_x)$ and $p(u_{xy})$ modeled in Eq. 3 if they are unobserved and change between source and target?

4. What justifies that the learned representations $Z_S$ and $Z_C$ truly follow the causal diagram in Fig. 1? Given the generative process of learning these representations (i.e. through $q(z_s \mid x)$ and $q(z_c \mid x)$), it could be argued that $Z_S$ and $Z_C$ are caused by $X$ rather than the other way around. Further, it is difficult to believe that a learned representation can contain more information about $Y$ than $X$, but this is what is implied by the graph (i.e. $Y$ and $X$ are independent given $Z_S$ and $Z_C$ but $Y$ is not independent of $Z_S$ and $Z_C$ given $X$?).

In addition, there are a few points that could use more elaboration:

5. At the beginning of Sec. 3.1, it is explained that the consideration of $U_x$ and $U_{xy}$ address two types of biases: selection bias and stereotype bias. This seems to be an interesting point and could be expanded.

6. Under Alg. 1, the paper mentions the necessity of assumptions to compensate for the lack of observations of $Z$ and $U$. These should be explicitly stated, as this seems to be the crux of the reasoning behind why the model works. Further, are some of these assumptions only relevant to certain types of data (e.g. images)?

I cannot recommend acceptance while I have these doubts, but I look forward to having them clarified in the authors’ responses.

**Questions:**

See weaknesses.

---

> ### Author Response · Authors · 2023-11-20
> **Responses to Reviewer ni4b (part 1)**
>
> **Invariance of $p(Y|X, do(Z_s))$:** Upon careful review, we acknowledge that our assertion regarding the variation of $p(U_{xy})$ was overstated. Our proposed interventional distribution, by setting specific values for $Z_s$ to mitigate the influence of the domain-specific variable $U_x$ on $Z_s$, effectively accounts for the invariant confounding effects and prevents the $U_x$ from influencing $Y$. Hence, it is invariant and transportable across domains. However, it is important to note that this framework may not generalize to new domains with different and unknown confounding effects.
>
> In light of this, we correct our initial assumption that the confounding effects, encompassing $p(U_{xy})$ and the causal mechanisms between $U_{xy}$, $Z_s$, and $Y$, remain consistent across training and test domains. This invariant confounding assumption aligns with standard practices widely adopted in works utilizing interventional distributions [1]. We will incorporate such a revision into the paper.
>
> Nevertheless, it's crucial to underscore that with the corrected assumption, our CIIRL remains innovative and has proven its efficacy across various benchmark distribution shift datasets:
> - 1) With the revised assumptions, our proposed interventional distribution maintains invariance and transportability, facilitating cross-domain inference.
> - 2) Our training procedure does not explicitly rely on the variation of $p(U_{xy})$. The distinct classes of $U$ can be associated with the diverse domain variable $U_x$. Our training procedure can still effectively distinguish the two types of representations.
> - 3) Empirical results affirm the existence of invariant latent confounders, as evidenced by the overall superior OOD prediction performance of CIIRL compared to predictions solely based on $Z_c$.
>
> > [1] Mao, Chengzhi, et al. "Causal transportability for visual recognition." Proceedings of the IEEE/CVF Conference on Computer Vision and Pattern Recognition. 2022.
>
>
> **The choice of $Z_s$:** Under our corrected assumption, the latent confounding effect remains invariant across domains. To infer the label for a test input $x^t$ using our interventional distribution, it is imperative to provide the true value of $Z_s$ for $x^t$ to accurately account for the confounding effects between $x^t$ and $y^t$. The distribution $p(z_s|x)$ utilized in Eq. (3) can be construed as a proxy distribution that enables us to derive the true values of $Z_s$ from an input $X$. The challenge lies in determining which learned distribution can be employed to approximate this proxy distribution. According to the SCM, we observe that $X$ is independent of $U$ given $Z=[Z_c, Z_s]$, rendering $p(X|Z)$ invariant and transportable. Therefore, a **reasonable and good choice** is to obtain the identifiable $z^t_s$ from the learned variational distribution $q(Z_s|X=x^t)$ with a high $p(X=x^t|Z_s=z^t_s)$. However, as $q(Z_s|X=x^t)$ represents a variational estimation of the desired proxy distribution, we average the interventional distribution over multiple samples of $z_s$ from $q(Z_s|X=x^t)$. In practice, randomly obtaining a $z_s$ value is likely to yield a low $p(X|Z)$ and contribute minimally to the calculation of the interventional distribution.
>
> **The modeling of $p(U_x)$ and $p(U_{xy})$:** Firstly, we revise our assumption by rectifying the notion that the distribution of $p(U_{xy})$ remains invariant across domains. In our context, $U_x$ denotes any information specific to the domain, with a common simplification being to assume that $U_x$ represents the domain index [2]. During the training procedure, we utilize a clustering algorithm to estimate the domain index for each training input. The objective of the training process is to acquire the encoder distributions $q(Z_s|X), q(Z_c|X)$ that produce a disentangled representation, which is identifiable and possesses an invariant generative mechanism $p(X|Z)$. The interventional distribution accommodates confounding effects and mitigates the influence of the domain variable $U_x$. Consequently, we can make inferences from the interventional distribution without knowledge of $U_{xy}$ and $U_x$ for the target domain.
>
> > [2] Lu, Chaochao, et al. "Invariant causal representation learning for out-of-distribution generalization." International Conference on Learning Representations. 2021.

---

> ### Author Response · Authors · 2023-11-20
> **Responses to Reviewer ni4b (part2)**
>
> **The SCM assumption:** We would like to emphasize that the causal mechanisms in the proposed SCM in Figure 1 are all assumptions that are widely adopted in the area of causal representation learning [1, 2], including the three following points:
> - 1) The latent high-level factors $Z$ can be separated into causal factors $Z_c$ and spurious factors $Z_s$.
> - 2) The input $X$ is generated by the the high-level factors $Z$.
> - 3) Causal factor $Z_c$ is either direct cause or effect of target $Y$.
>
> We cannot prove these assumptions always hold in real-world data and we admit the effectiveness and soundness of our derived theorem and algorithm is built upon these assumptions. We believe the learned representation $Z_s$ and $Z_c$ are the factors that satisfy the causal graph by
> - 1) parameterizing the joint distribution regarding all the variables of interest into conditional distributions adhering to the causal mechanisms in Figure 1;
> - 2) establishing the identifiability of the learned $Z_s, Z_c$.
>
> We appreciate the suggestion of further elaboration of these two types of biases and will revise the paper accordingly.
>
> We acknowledge the recommendation to underscore the importance of assumptions and will incorporate this emphasis into our paper accordingly. It's worth noting that our method is not confined solely to image data. The validity of our theorem and the effectiveness of the algorithm persist as long as the assumptions we posit apply to the given data. As an illustration, our approach can be extended to encompass text data, demonstrating the versatility of our framework.

---

### Official Review · Reviewer_RUwh · 2023-11-03

**Soundness:** 2 fair
**Presentation:** 3 good
**Contribution:** 2 fair
**Rating:** 3
**Confidence:** 4

**Summary:**

In this paper, the authors investigate the problem of domain generalization, where the target domain datasets are unobserved during the training phases. To solve this problem, the authors propose a structural causal model with latent variables to model the causal mechanism. Sequentially, the authors conduct an intervention on the spurious representations to remove the spurious correlations and further learn the invariant interventional distribution. The authors evaluate the proposed methods on several datasets and achieve ideal performance.

**Strengths:**

1.	The authors leverage the causal knowledge to address the domain generalization problem.
2.	The authors evaluate the proposed methods on several datasets.

**Weaknesses:**

1.	One important issue is the confusedness of the type of variables in Figure 1. In the domain generalization task, the domain labels are usually observed. However, it is unclear if $U_x$ and $U_{x,y}$ are observed variables or not.
2.	Moreover, the authors mentioned that $P^S(Y|X,do(Z_S))= P^T(Y|X,do(Z_S))$ according to Figure 2(b). But if $U_{x,y}$ is influenced by different domains, the aforementioned equation is not true.
3.	The proposed causal generation process is similar to that of [1], it is suggested that the authors should provide a discussion between the proposed causal generation process and [1]. Moreover, it seems to be impossible to conduct do-calculus on the latent variables without identification guarantees of the latent variables.

[1] Partial disentanglement for domain adaptation  Lingjing Kong, Shaoan Xie, Weiran Yao, Yujia Zheng, Guangyi Chen, Petar Stojanov, Victor Akinwande, Kun Zhang Proceedings of the 39th International Conference on Machine Learning, PMLR 162:11455-11472, 2022.

**Questions:**

N.A.

---

> ### Author Response · Authors · 2023-11-20
> **Responses for Reviewer RUwh**
>
> **Latent $U_x$ and $U_{xy}$:** Our approach is specifically designed to enhance out-of-distribution (OOD) prediction in situations where the domain variable $U_x$ and confounder $U_{xy}$ are not known. While domain indices are provided for specific tasks/datasets like PACS and VLCS, acquiring them for general real-world tasks poses significant challenges.
>
> **Invariance of $p(Y|X, do(Z_s))$:** Upon careful review, we acknowledge that our assertion regarding the variation of $p(U_{xy})$ was overstated. Our proposed interventional distribution, by setting specific values for $Z_s$ to mitigate the influence of the domain-specific variable $U_x$ on $Z_s$, effectively accounts for the invariant confounding effects and prevents the $U_x$ from influencing $Y$. Hence, it is invariant and transportable across domains. However, it is important to note that this framework may not generalize to new domains with different and unknown confounding effects.
>
> In light of this, we correct our initial assumption that the confounding effects, encompassing $p(U_{xy})$ and the causal mechanisms between $U_{xy}$, $Z_s$, and $Y$, remain consistent across training and test domains. This invariant confounding assumption aligns with standard practices widely adopted in works utilizing interventional distributions [1]. We will incorporate such a revision into the paper.
>
> Nevertheless, it's crucial to underscore that with the corrected assumption, our CIIRL remains innovative and has proven its efficacy across various benchmark distribution shift datasets: 1) With the revised assumptions, our proposed interventional distribution maintains invariance and transportability, facilitating cross-domain inference. 2) Our training procedure does not explicitly rely on the variation of $p(U_{xy})$. The distinct classes of $U$ can be associated with the diverse domain variable $U_x$. Our training procedure can still effectively distinguish the two types of representations. 3) Empirical results affirm the existence of invariant latent confounders, as evidenced by the overall superior OOD prediction performance of CIIRL compared to predictions solely based on $Z_c$.
>
> > [1] Mao, Chengzhi, et al. "Causal transportability for visual recognition." Proceedings of the IEEE/CVF Conference on Computer Vision and Pattern Recognition. 2022.
>
> **Comparison to [2]:**  We appreciate that the reviewer recommends this relevant paper. The casual graph for the data generation process closely resembles our SCM. Specifically, our causal graph shares similarities with the one presented in [2] in the following aspects:
> - 1) we both separate the latent representation into (invariant) causal representation $Z_c$ and (variant) spurious representation $Z_s$. The input $X$ is generated by both types of representation.
> - 2) The spurious representation varies from domain to domain since it is controlled by a domain-specific variable $U_x$.
> - 3) There exists a high-level invariance confounder between $Z_s$ and $Y$ ($\tilde{Z}_s$ in their graph).
>
> However, the graph in [2] assumes that causal features as the parent variables to target while we use the child variables.
>
> Algorithmically, both our method and iMSDA from [2] address the confounding issue. iMSDA utilizes domain index information to estimate the confounder, whereas we focus on constructing the interventional distribution. Furthermore, iMSDA is tailored for domain adaptation tasks and necessitates access to test domain data during training. Given these distinctions, a direct comparison between our method and iMSDA would not be equitable.
>
> > [2] Partial disentanglement for domain adaptation Lingjing Kong, Shaoan Xie, Weiran Yao, Yujia Zheng, Guangyi Chen, Petar Stojanov, Victor Akinwande, Kun Zhang Proceedings of the 39th International Conference on Machine Learning, PMLR 162:11455-11472, 2022.
>
> **Identifiability of representation $Z_c, Z_s$:** We strongly agree that establishing the identifiability of the latent variables $Z$ is crucial for performing do-calculus on them. Therefore, we leverage the theoretical results presented by Kivva et al. (2022) to demonstrate the identifiability of the $Z$ obtained through our learning framework. Please refer to the paragraph below the algorithm in Section 3.2 for more details.
>
> > [3] Kivva, Bohdan, et al. "Identifiability of deep generative models without auxiliary information." Advances in Neural Information Processing Systems 35 (2022): 15687-15701.

---

### Meta-Review · Area_Chair_fCi4 · 2023-12-04

**Metareview:**

The paper deals with domain generalization, where the target domain datasets are unobserved. This work intervenes on spurious representations to remove correlations and learn an invariant distribution. Their method performs well across various datasets, essentially focusing on learning causal and spurious representations to guide predictions based on an invariance relation.

All four reviews are toward rejection with ratings of 3, 3, 5, and 5 with confidence of 4, 4, 3, and 3 respectively. The issues raised by the reviews are critical including the theoretical plausibility (RUwh, kccw, Rf1b), and the clarity (ni4b).

**Justification For Why Not Higher Score:**

The issues of theoretical plausibility and clarity.

**Justification For Why Not Lower Score:**

N/A

---

### Decision · Program_Chairs · 2024-01-16

Reject